# Conditional deletion of glucocorticoid receptors in rat brain results in sex-specific deficits in fear and coping behaviors

Jessie R Scheimann[1†], Rachel D Moloney[1†*], Parinaz Mahbod[1], Rachel L Morano[1], Maureen Fitzgerald[1], Olivia Hoskins[1], Benjamin A Packard[1], Evelin M Cotella[1], Yueh-Chiang Hu[2,3,4], James P Herman[1*]

[1]Department Pharmacology and Systems Physiology, University of Cincinnati, Cincinnati, United States; [2]Division of Developmental Biology, Cincinnati Children's Hospital Medical Center, Cincinnati, United States; [3]Department of Pediatrics, University of Cincinnati College of Medicine, Cincinnati, United States; [4]Division of Reproductive Sciences, Cincinnati Children's Hospital Medical Center, Cincinnati, United States

**\*For correspondence:**
Rachel.Moloney@uc.edu (RDM);
James.Herman@uc.edu (JPH)

[†]These authors contributed equally to this work

**Competing interests:** The authors declare that no competing interests exist.

**Abstract** Glucocorticoid receptors (GR) have diverse functions relevant to maintenance of homeostasis and adaptation to environmental challenges. Understanding the importance of tissue-specific GR function in physiology and behavior has been hampered by near-ubiquitous localization in brain and body. Here we use CRISPR/Cas9 gene editing to create a conditional *GR* knockdown in Sprague Dawley rats. To test the impact of cell- and region-specific *GR* knockdown on physiology and behavior, we targeted *GR* knockdown to output neurons of the prelimbic cortex. Prelimbic knockdown of *GR* in females caused deficits in acquisition and extinction of fear memory during auditory fear conditioning, whereas males exhibited enhanced active-coping behavior during forced swim. Our data support the utility of this conditional knockdown rat to afford high-precision knockdown of *GR* across a variety of contexts, ranging from neuronal depletion to circuit-wide manipulations, leveraging the behavioral tractability and enhanced brain size of the rat as a model organism.
DOI: https://doi.org/10.7554/eLife.44672.001

## Introduction

The glucocorticoid receptor (GR) is a ligand (glucocorticoid)-activated transcription factor that participates widely in functions related to homeostasis and adaptation (*Herman, 1993*). Functional properties of GR action have been widely queried, using pharmacological, viral vector and electrophysiological approaches, with most of the literature using rat models (*McKlveen et al., 2013*; *McKlveen et al., 2016*; *Ghosal et al., 2014*; *Solomon et al., 2014*; *Wulsin et al., 2010*). Recent decades have seen an increase in the use of mouse models to provide Cre-driver mediated deletion in specified cell populations (e.g. CaMKIIα neurons of the forebrain, Simpleminde-1 (Sim-1) neurons of the hypothalamus; *Nahar et al., 2015*; *Solomon et al., 2015*). While yielding valuable information on cell type-specific GR actions (*Solomon et al., 2012*), the use of mice has proven problematic with regard to high-resolution behavioral and physiological analyses due to the different behavioral repertoire of mice (high strain and inter-experiment variability in memory tests) and the small size (e.g., prohibits blood sampling of large volumes) (*Whishaw and Tomie, 1996*; *Parker et al., 2014*; *Ellenbroek and Youn, 2016*; *Colacicco et al., 2002*).

The emergence of CRISPR (Clustered Regularly Interspaced Short Palindromic Repeats)/Cas 9-mediated gene editing now affords gene targeting in numerous species, including rat (*Wang et al., 2016*; *Shao et al., 2014*; *Ran et al., 2013*). For example, CRISPR/Cas9 methods allow for introduction of exon-flanking LoxP sites to generate conditional knockdown alleles and subsequent gene deletion following exposure to Cre recombinase using either driver lines or viral vector approaches. In this study, we used CRISPR/cas9 to specifically insert LoxP sites to sequences flanking exon 3 of the *Nr3c1* (*GR*) gene in Sprague Dawley rats (an outbred strain commonly used to explore GR function) via homology-directed repair (HDR). Our data provide validation of the *Nr3c1* gene editing in rats, and the use of viral vector-mediated Cre delivery to demonstrate targeting of GR knockdown to specific cell types, brain regions and circuits. Functional efficacy of GR knockdown in the prelimbic division (PL) of the medial prefrontal cortex (PL-PFC) was verified by sex-specific deficits in extinction of conditioned fear in females and a shift to active coping in the forced swim test (FST) in males. Overall, our study shows the utility of this gene editing technique to generate conditional gene deletion models that can leverage the considerable advantage of rats in behavioral and physiological research.

## Results

We directly manipulated the genome of Sprague Dawley rat zygotes by CRISPR/Cas9 to generate the floxed *Nr3c1* allele. We used a dual sgRNA strategy to delete the sequence containing exon 3 of the *Nr3c1* gene and repaired it with a cutting resistant donor plasmid that contains the deleted sequence, two flanking LoxP sites at the cut location of the sgRNA recognition sites, a right homologous arm at 2.57 kb, and a left homologous arm at 1.95 kb (*Figure 1A and B*). Because truncated sgRNAs increase targeting specificity (*Fu et al., 2014*), we chose six sgRNAs with various lengths (17–20 nt) (*Appendix 1—table 1*) and validated their editing activity in rat C6 glioma cells by T7E1 assay (*Appendix 1—figure 1*). We picked two sgRNAs, sg-2 and sg-6, for targeting the 5' and 3' sequences of exon 3, respectively. Both sgRNAs were 17nt in length, which is expected to provide high specificity (*Fu et al., 2014*). The two selected sgRNAs, Cas9 mRNA, and the donor plasmid were microinjected into ~60 rat zygotes, followed by embryo transfer into pseudopregnant female rats. Seventeen pups were born. We identified that one of them (No. #60) was correctly targeted, which was confirmed by PCR with the external primers (*Figure 1C and D*) paired with the primers partially containing LoxP sequences (P5-P6 and P10-P8 for 5' and 3' ends, respectively; *Figure 1E and F*). It was further confirmed by primer pairs P5-P7 and P9-P8, followed by ClaI and BamHI enzyme digestion and Sanger sequencing (data not shown), and by additional Sanger sequencing of the entire targeted area between P5 and P8. This allele contains an A/T base change at chr18:31,744,353, compared to the rn6 reference genome, which is located in a non-conserved intronic region and unlikely to be of functional consequence. We verified copy number and used plasmid backbone PCR to exclude the possibility of random integration of the donor plasmid (*Appendix 1—figure 2*). We have consistently bred the offspring of rat #60 to homozygosity (*Figure 1G*).

### Validation of conditional GR knockdown: Viral vector targeting

To test the efficacy of this novel rat line, we administered adenoviral Cre recombinase constructs to drive regional, cell type-specific and projection-specific knockdown of GR. Regional knockdown targeted the basolateral amygdala (BLA), using human synapsin promoter-driven Cre recombinase (AAV8-hSyn-Cre, UNC Vector Core, NC, USA) microinjections. Cre$^+$ cells were devoid of nuclear GR immunoreactivity in SD:nr3c1$^{fl/fl}$ rats (*Figure 2A*), whereas the vast majority of Cre$^+$ cells co-expressed GR in wildtype controls (SD:nr3c1$^{wt}$) (*Figure 2B*) injected with the same viral construct (AAV8-hSyn-Cre, UNC Vector Core).

We then assessed cell-type specific knockdown by injection of AAV9.CamKII.HI.eGFP-Cre.WPRE.SV40 (CaMKIIα Cre) virus into the PFC of SD:nr3c1$^{wt}$ or SD:nr3c1$^{fl/fl}$ rats. Infusions of CaMKIIα Cre caused widespread loss of GR immunoreactivity within GFP$^+$cells in the PFC of CaMKIIα Cre virus injected SD:nr3c1$^{fl/fl}$ rats (*Figure 2C* yellow arrows), but no GR loss is noted in wildtype control (SD:nr3c1$^{wt}$) injected rats (*Figure 2D* white arrows) or in uninfected CaMKIIα cells in SD:nr3c1$^{fl/fl}$ rats (*Figure 2C* white arrows).

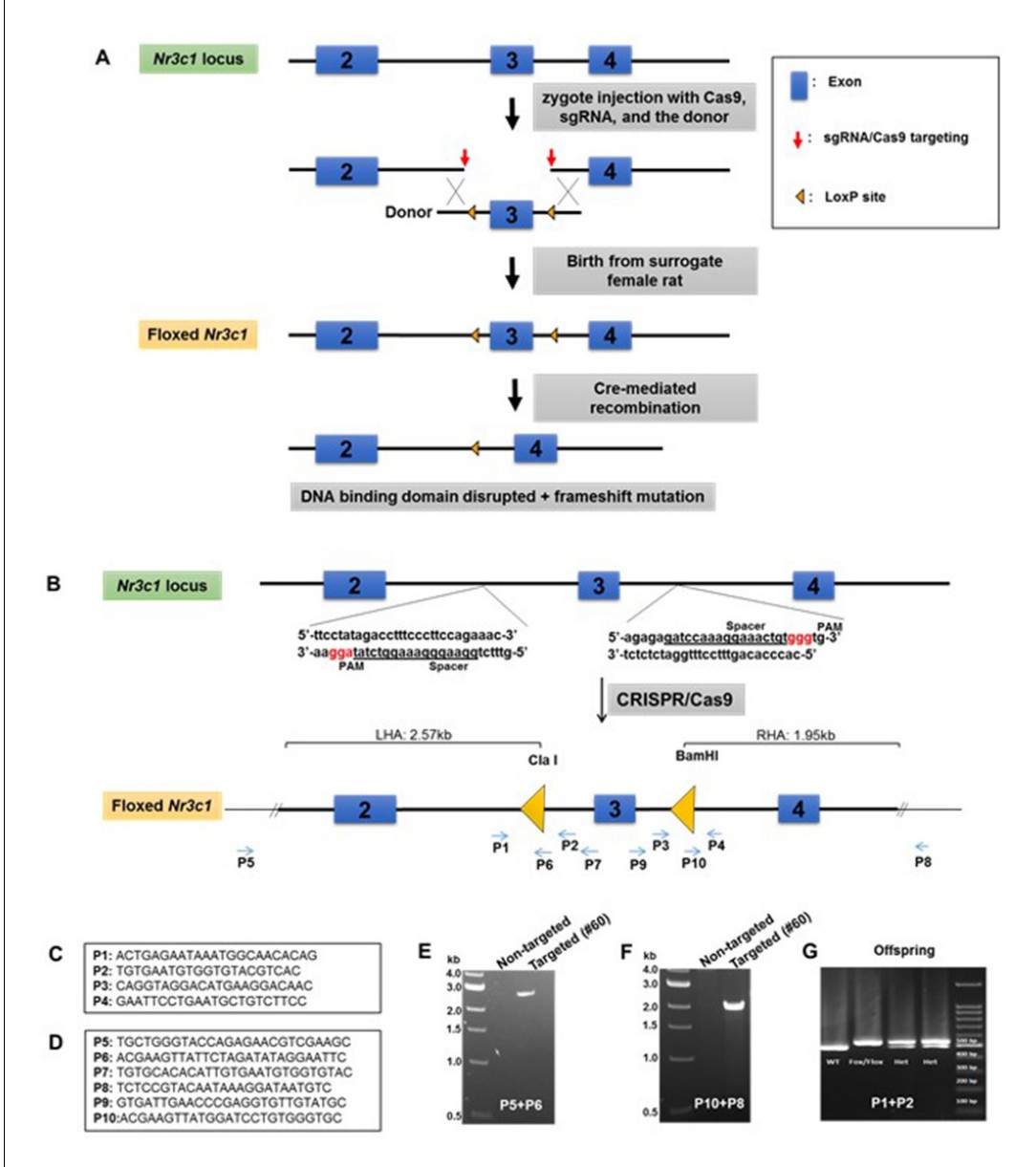

**Figure 1.** Generation of *Nr3c1* conditional knockdown rat. (**A**) Schematic illustration of the targeting strategy. Two LoxP sequences flanking exon three were inserted via CRISPR/Cas9-mediated deletion by two sgRNAs, followed by homology-directed repair with a donor plasmid. The CRISPR reagents were injected into the cytoplasm of rat zygotes, followed by embryo transfer into surrogate mothers for fetal development and birth. Correctly targeted rats were bred to homozygosity. Deletion of exon three is achieved by recombination of two LoxP sequences upon the exposure of Cre recombinase. (**B**) The donor plasmid contains two LoxP sequences flanking exon 3, restriction enzyme sites, and two homologous arms. The LoxP sequence is located at the sgRNA cut site (three nucleotides prior to the PAM) to block re-cutting. (**C**) Primers used to confirm the correctly targeted events are listed in the boxes. (**D**) Primers P5 and P8 are external to homologous arms. (**E–G**) Sample PCR results are shown.

DOI: https://doi.org/10.7554/eLife.44672.002

Finally, we used an intersectional approach to examine connectional knockdown of GR, focusing on PFC projection neurons to the BLA. Retrogradely-infected (Cre[+]) neurons were observed in the PFC after administration of AAVrg pmSyn1-EBFP-Cre, (Addgene) to the BLA. We did not observe GR immunoreactivity in SD:nr3c1$^{fl/fl}$ rats (*Figure 2E*), whereas substantial proportions of PFC-BLA projecting neurons contained GR immunoreactivity in control virus (AAVrg-CAG-GFP, Addgene) injected animals (*Figure 2F*). Although we made every effort to employ appropriate viral controls, we were limited by the availability of viral constructs, specifically AAV retrograde constructs, and this

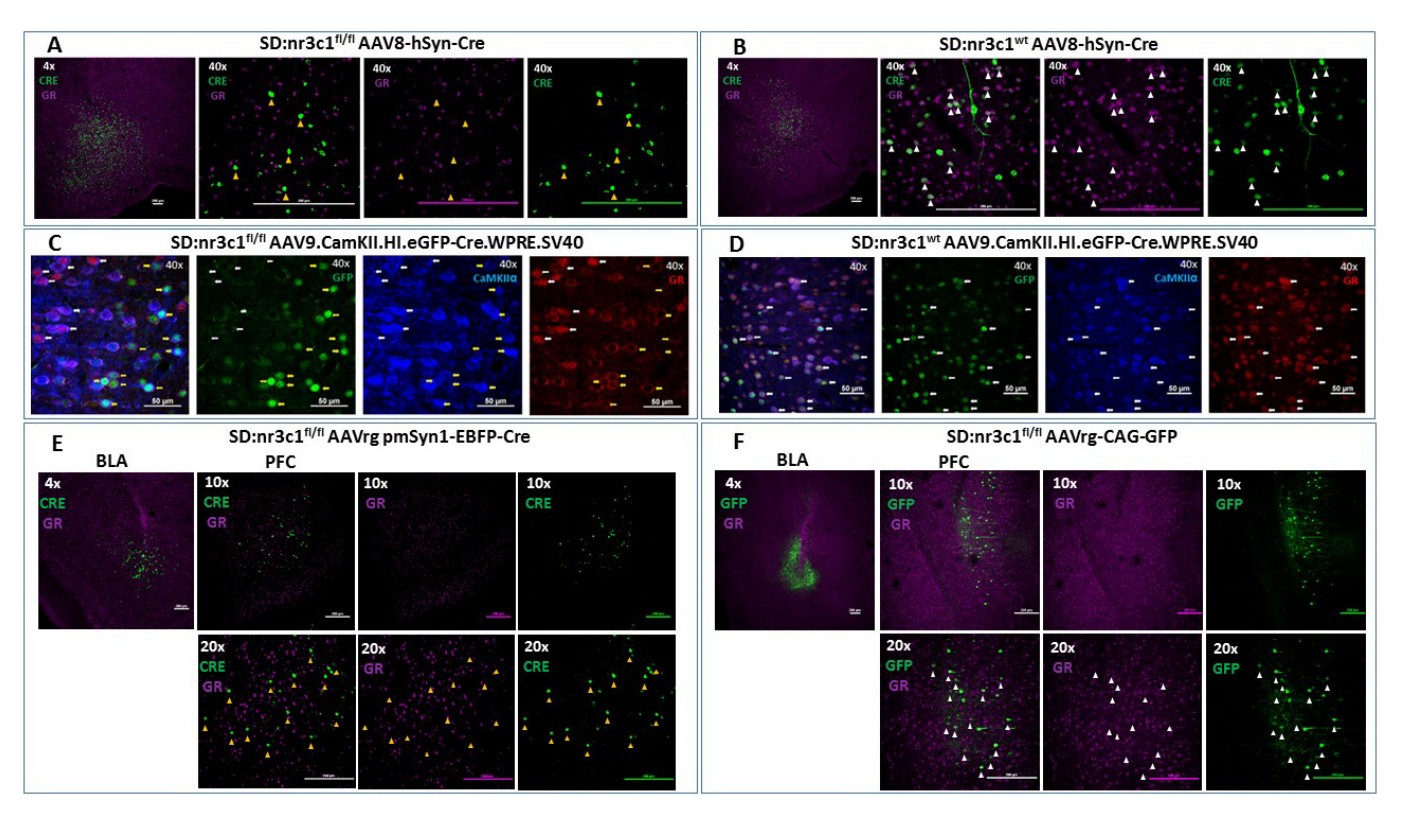

**Figure 2.** Viral validation of conditional glucocorticoid receptor (GR) knockdown rat. (**A**) AAV8-hSyn-Cre administration to the basolateral amygdala (BLA) in SD:nr3c1$^{fl/fl}$ rats (panel A1- site of injection, 4X). Note the absence of GR (purple) in Cre$^+$ neurons (green) (panels A2-4, 40x, yellow triangles). (**B**) AAV8-hSyn-Cre administration to the BLA in SD:nr3c1$^{wt}$ rats (panel B1- site of injection, 4X). Note the presence of GR (purple) in Cre$^+$ neurons (green) (panels B2-4, 40x, white triangles). (**C**) AAV9.CamKII.HI.eGFP-Cre.WPRE.SV40 (CaMKIIα Cre) injected into SD:nr3c1$^{f/f}$ rats results in decreased GR (C4 yellow arrows) in cells infected with virus as shown by GFP labeling (C2 yellow arrows) that are also CaMKIIα positive (C3 yellow arrows). CaMKIIα cells not infected with GFP, show endogenous GR expression (C1-4 white arrows). (**D**) CaMKIIα Cre injected into SD:nr3c1$^{wt}$ rats (panel C2-4 40x white arrows) shows endogenous GR staining (D4) in cells infected with virus shown in GFP (D2) and CaMKIIα positive (D3). (**E**) AAVrg-pmSyn1-EBFP-Cre administration to the BLA in SD:nr3c1$^{fl/fl}$ rats (panel E-1 site of injection, 4X) and retrograde trafficking of the virus to cell somas in the prefrontal cortex (PFC) (panel E2-4). Note absence of GR expression (purple) in Cre$^+$ neurons (green) (panel E5-7, 20x, yellow triangles). (**F**) AAVrg-CAG-GFP administration to the BLA in SD:nr3c1$^{fl/fl}$ rats (panel F1- site of injection, 4X) and retrograde trafficking of the virus to cell somas in the PFC (panel F2-4, 10X). Note expression of GR (purple) in GFP$^+$ neurons (green) (panel F5-7, 20x, white triangles).

DOI: https://doi.org/10.7554/eLife.44672.003

is a caveat of the current study. For our intersectional approach we used the same AAV serotype (AAVrg) for both knockdown and controls however, the promoter (pmSyn1 vs CAG) and fluorescent reporter (EBFP vs GFP) were different. These constructs were chosen as the best available at the time of running the experiments. These studies highlight the novel use of this rat model to query not only the role of GR in a specific region in isolation but also how GR functions as part of an integrated circuit.

## Behavioral consequences of targeted GR knockdown; Implications for fear and coping behaviors

We next performed a functional test of viral Cre-mediated GR knockdown, focusing on the PL-PFC. SD:nr3c1$^{fl/fl}$ (GRKD) rats and wild type littermate controls SD:nr3c1$^{wt}$ (Control) all received injection of AAV9.CamKII.HI.eGFP-Cre.WPRE.SV40 (CaMKIIα Cre) [Penn Vector Core] into the PL-PFC *Figure 3E*. CaMKIIα is a calcium binding protein that is most commonly found in glutamatergic neurons of the forebrain (although there is evidence of the presence of CaMKIIα in some GABAergic interneurons in the amygdala and striatum and possibly the PFC; *Jennings et al., 2013*; *Klug et al.,*

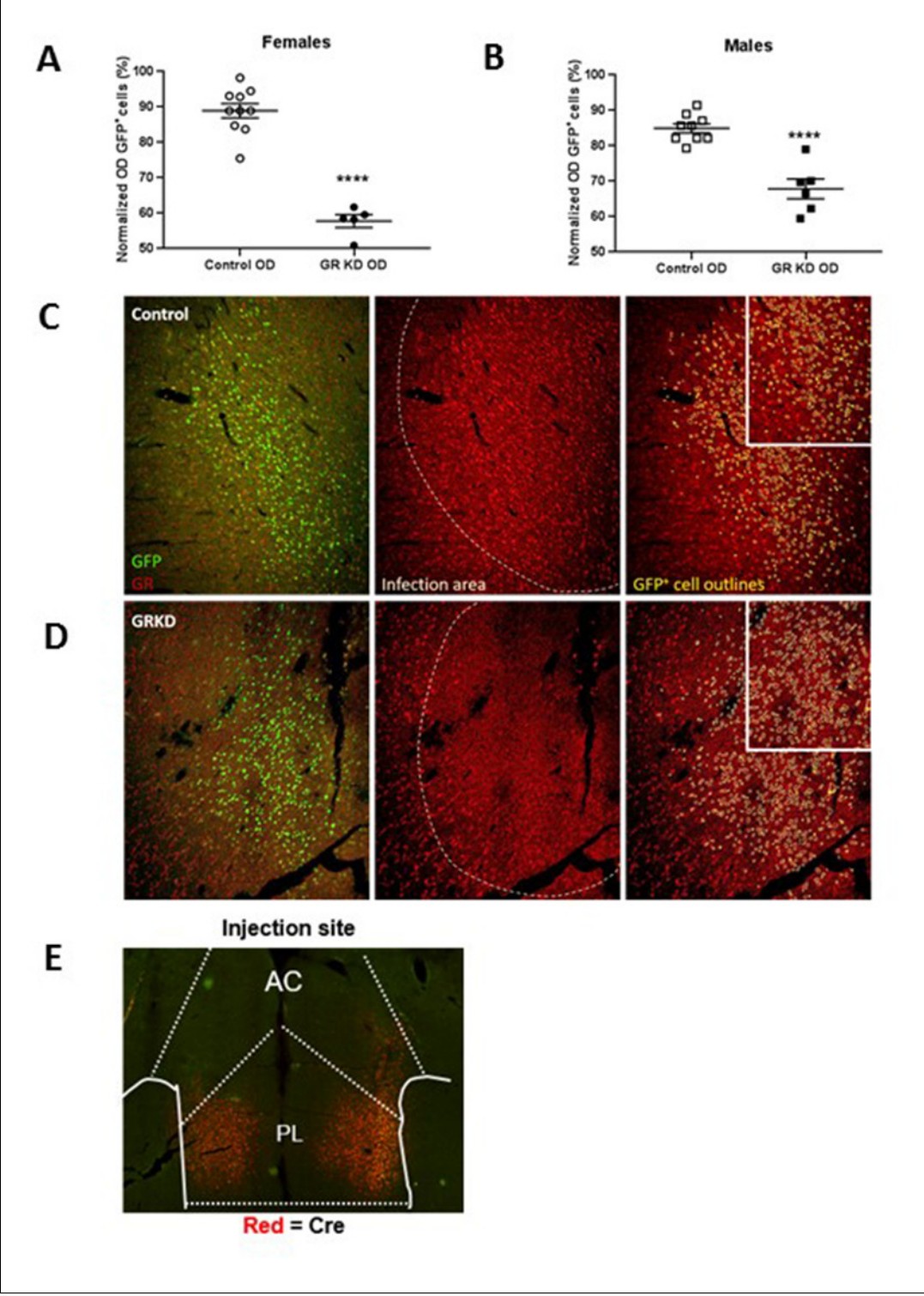

**Figure 3.** Verification of glucocorticoid receptor (GR) knockdown in the prelimbic division of the prefrontal cortex (PL-PFC). (A) Female SD:nr3c1$^{fl/fl}$ rats injected with AAV9.CamKII.HI.eGFP-Cre.WPRE.SV40 show reduced GR expression in virus infected neurons (GFP$^{+}$) compared to controls (****p<0.0001, unpaired t-test, GRKD = SD: nr3c1$^{fl/fl}$ plus CaMKIIα Cre, n = 5; Control = SD:nr3c1$^{wt}$ plus CaMKIIα Cre, n = 10). (B) Male SD:nr3c1$^{fl/fl}$ rats injected with AAV9.CamKII.HI.eGFP-Cre.WPRE.SV40 show reduced GR expression in virus infected neurons (GFP$^{+}$) compared to controls (****p<0.0001, unpaired t-test, GRKD = SD:nr3c1$^{fl/fl}$ plus CaMKIIα Cre, n = 6; Control = SD: nr3c1$^{wt}$ plus CaMKIIα Cre, n = 9). (C) Representative image of GR expression in GFP+ neurons in the area of
*Figure 3 continued on next page*

*Figure 3 continued*
infection (PL-PFC) in a SD:nr3c1$^{wt}$ rat. Green = viral infected neurons, Red = GR, yellow = GFP cell outline. (**D**) Representative image of GR knockdown in GFP+ neurons in the area of infection (PL-PFC) in a SD:nr3c1$^{fl/fl}$ rat injected with AAV9.CamKII.HI.eGFP-Cre.WPRE.SV40. Green = viral infected neurons, Red = GR, yellow = GFP cell outline. (**E**) Representative image of CaMKIIα injection site and area of infection in the PL-PFC. Red = Cre recombinase protein.
DOI: https://doi.org/10.7554/eLife.44672.004

*2012*; *Robison et al., 2014*), thus we are targeting a glutamatergic enriched populations of neurons in the PL-PFC.

We first confirmed GR knockdown in virus infected neurons in both females and males using immunohistochemistry [Females = T(13)=9.752; p<0.0001 (*Figure 3A*)]; [Males = T(13)=6.202; p<0.0001 (*Figure 3B*)].

The PFC plays a critical role in extinction of emotional memory (e.g. conditioned fear), selection of emotional coping strategy, and hypothalamic pituitary adrenal (HPA) axis reactivity (*Amat et al., 2005*; *Giustino and Maren, 2015*; *Quirk et al., 2000*; *Quirk et al., 2006*). We therefore tested whether GR knockdown in CaMKIIα targeted neurons in this region affected extinction of conditioned fear to an auditory fear conditioning paradigm, and behavioral coping during the FST. For fear conditioning, rats were exposed to five tone shock pairings on the first day (acquisition), followed by 2 days of 20 tones without a paired shock, (extinction and extinction recall). Data was binned for clarity into five tones per bin. Freezing during the tones was measured as fear behavior. Freezing in female rats with PL-PFC targeted GRKD (SD:nr3c1$^{fl/fl}$ plus CaMKIIα Cre) increased in acquisition [GRKD F(1,52) = 5.553; p=0.035], but there was no significant interaction effect [GRKD x tone F(4,52) = 1.292; p=0.285] (*Figure 4A*). Extinction of fear conditioning was delayed in the female GRKD group relative to controls, as indicated by a significant time x GRKD interaction effect [GRKD x time F(3,39) = 4.184; p=0.012] (*Figure 4A*). Extinction recall was also impaired in female PL-PFC GRKD rats relative to controls (significant time x GRKD interaction) [GRKD x time F (3,31)=3.796; p=0.020] (*Figure 4A*). Meanwhile, males with PL-PFC targeted GRKD (SD:nr3c1$^{fl/fl}$ plus CaMKIIα Cre) did not differ from controls [Acquisition GRKD x tone F(1,79)=1.827; p=0.136] [Extinction GRKD x tone F(1,63) = 0.873; p=0.463] [Extinction Recall GRKD x tone F(1,63) = 0.592; p=0.624] (*Figure 4B*). While most lesion and inactivation studies have shown that the PL-PFC is critical for appropriate fear responding (*Giustino and Maren, 2015*; *Quirk et al., 2000*; *Quirk et al., 2006*), the use of the GR floxed rat model has shown that GR in the CaMKIIα neurons of the PL-PFC may be more critical for female expression and extinction of conditioned fear, and less so for males.

The PFC has also been implicated in coping behaviors and behavioral adaptation during acute and chronic stress. We used the FST to investigate remodeling of behavioral coping strategy during an acute stress challenge. There was no effect of PL-PFC-driven GRKD (SD:nr3c1$^{fl/fl}$ plus CaMKIIα Cre) on immobility time in the FST on either day 1 or day 2 for male rats [Day 1 immobility T(14) = −1.848; p=0.085] (*Figure 4C*) [Day 2 immobility T(14) = −1.411; p=0.180] (*Figure 4D*). However, males with CaMKIIα-driven PL-PFC GRKD (SD:nr3c1$^{fl/fl}$ plus CaMKIIα Cre) increased diving frequency on day 1 [T(14)=3.252; p=0.005] (*Figure 4E*) relative to controls, consistent with an altered coping strategy. This behavior persisted into the second day of the FST test, with significantly higher number of dives for males with PL-PFC GRKD (SD:nr3c1$^{fl/fl}$ plus CaMKIIα Cre) [T(14) = - 3.468; p=0.003] (*Figure 4F*). Females with PL-PFC GRKD (SD:nr3c1$^{fl/fl}$ plus CaMKIIα Cre) did not differ from controls in total immobility (*Figure 4G–H*) or number of dives (*Figure 4I–J*) [Day 1 Immobility, T(13) = 0.077; p=0.939] [Day 2 immobility, T(13) = −1.083; p=0.298] [Day 1 dives, T(13) = −0.187; p=0.854] [Day 2 dives, T(13) = −1.316; p=0.211]. Although there were no changes in total immobility in the FST, increased diving in the males with PL-PFC GRKD suggests GR in CaMKIIα neurons of the PL-PFC may facilitate a shift, selectively in males, to more diverse active escape behaviors than just swimming and climbing alone.

Our lab has previously shown that GRKD in the PL-PFC increases corticosterone to acute restraint (*McKlveen et al., 2013*). We subjected male and female GRKD and control rats to a 30min acute restraint challenge and took tail blood samples at 0 min, 15 min, 30 min, 60 min, and 120 min from the start of restraint to measure plasma corticosterone. There was a GRKD x time interaction [F(4,45) =3.020; p=0.027] in females, with post hoc analysis revealing delayed shut-off (120 min time point)

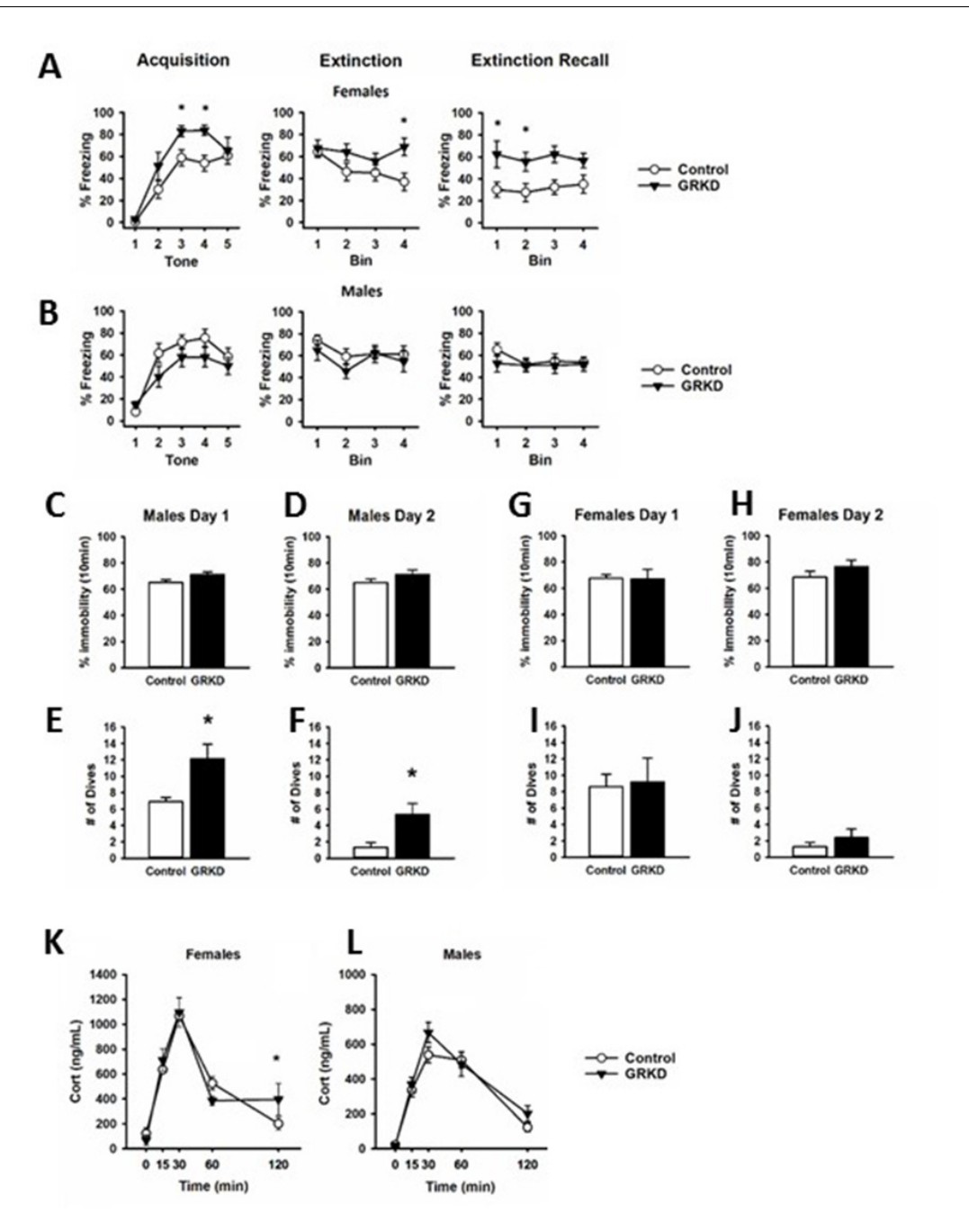

**Figure 4.** Behavioral profile of glucocorticoid receptor (GR) knockdown in CaMKIIα cells in the prelimbic division of the prefrontal cortex (PL-PFC). **(A)** Female rats with PL-PFC CaMKIIα GRKD showed heightened freezing to tone shock pairings compared to controls, as well as deficits in fear extinction, and heightened freezing during extinction retrieval indicating an inability to extinguish conditioned fear (*=$p < 0.05$ with Bonferroni posthoc). **(B)** Male rats with PL-PFC CaMKIIα GRKD did not differ from controls during auditory fear conditioning. **(C–D)** Male rats with CaMKIIα GRKD in the PL-PFC did not show differences from controls in immobility on either day in the forced swim test, **(E–F)** but PL-PFC CaMKIIα GRKD male rats did show a greater number of dives on both days suggesting a more active coping behavior (*=$p < 0.05$ Student's two-tailed T-test). **(G–J)** Female rats with PL-PFC CaMKIIα GRKD did not show differences from controls in the forced swim test. **(K)** Females with PL-PFC CaMKIIα GRKD show a significant GRKD x time interaction in corticosterone after acute restraint as well as higher corticosterone at the 120 min time point ($p=0.050$ with Bonferroni posthoc). **(L)** Male rats with PL-PFC CaMKIIα GR knockdown do not show differences from control in corticosterone release to acute restraint. (GRKD = SD:nr3c1[fl/fl] plus AAV CaMKIIα Cre, n = 5–6 male and female; Control = SD:nr3c1[wt] plus AAV CaMKIIα Cre, n = 10 male and female).
DOI: https://doi.org/10.7554/eLife.44672.005

of the HPA axis stress response in PL-PFC-GRKD rats [p=0.050] (*Figure 4K*). There was no effect of GRKD [F(1,44)=2.046; p=0.180] or GRKD x time interaction observed in males [F(4,44)=0.935; p=0.452] (*Figure 4L*). The data suggest that female SD:nr3c1$^{fl/fl}$ rats with PL-PFC GRKD may have deficits in negative feedback inhibition due to the inability to return to a similar resting level of corticosterone as control rats. The lack of differences after PL-PFC GRKD in the males compared to our previous study may be due to the specificity of our virus to knockdown primarily in CaMKIIα cells, whereas previously a ubiquitous promotor on a lentiviral delivered shRNA was used which targets all neuronal cell types, including inhibitory GABAergic interneurons (*McKlveen et al., 2013*).

## Discussion

Prior studies document generation of knockdown and Cre dependent knockdown rat models using CRISPR/Cas9 as a molecular tool (*Shao et al., 2014*; *Ma et al., 2014*) reviewed in *Wang et al. (2016)*. Here we created a conditional (Cre-recombinase dependent) GR knockdown rat using CRISPR/Cas9. Validation of genome integration was accomplished by PCR, and Cre-dependent knockdown by site and circuit specific viral vector expression of Cre-recombinase. Finally, efficacy and consequences of GR knockdown in the PL-PFC were supported by sex-specific effects in behavior. The CaMKIIα promoter is thought to largely direct Cre expression to cortical projection neurons with over 80% specificity and efficacy (*Wood et al., 2019*) and thus it is likely our manipulations were able to successfully target a cell population composed mostly of excitatory projection neurons.

Deficits in fear conditioning and alterations in behavioral coping style are a common phenotype in murine GR knockdown models as well as following chronic stress (*McKlveen et al., 2013*). We used site and cell specific knockdown of GR to demonstrate that CaMKIIα cells in the PL-PFC require GR for appropriate fear responses and extinction in females. Conversely, male rats lacking GR in the PL-PFC, shift to an active escape behavior in the FST, suggesting adoption of an active coping strategy. Females (but not males) evidenced deficient shut-off of the HPA axis stress response, consistent with a sex-specific dependence of PL-PFC GR for full feedback inhibition. The data highlight a strong interaction between GR signaling and sex in coordination of prefrontal cortical signaling mechanisms. Moreover, the data provide functional evidence that CRISPR/Cas9 can be used to provide high-resolution cell and site specific assessment of GR action in brain, leveraging the advantages of the rat as a model organism.

Prior studies have used promoter-specific Cre driver mouse lines to generate targeted GR knockdown in mice. Mouse studies demonstrated that CaMKIIα-Cre directed GR deletion in the forebrain (targeting exon 2 or 3) promoted anxiety-related behavior, passive coping, and corticosterone hypersecretion in male, but not female mice (*Solomon et al., 2012*; *Boyle et al., 2006*; *Hartmann et al., 2017*). It is important to note that the mouse CaMKIIα driver line deletes GR in multiple brain regions, including cortex, hippocampus, BLA, caudate, and bed nucleus of the stria terminalis, many of which interface with stress and emotional behavior. The extensive knockdown makes it impossible to specify circuit-specific roles of stress hormone signaling in behavior and stress physiology. Prior studies have not specified GR-specific deficits in cognitive behaviors, which can be difficult to assess in mouse models (*Whishaw and Tomie, 1996*; *Parker et al., 2014*; *Ellenbroek and Youn, 2016*). Here, we show that by using these viral constructs with specific promoters (CaMKIIα) in GR floxed rats, we can not only drive expression of Cre-recombinase in a cell-type specific manner, but also a defined region, allowing precise investigation of the role of GR in behavior.

There is a possibility of genetic modifications, even with the highly site specific editing of the CRISPR/Cas9 system, when inserting LoxP sites into the genome (reviewed in *Zhang et al., 2015*; *Fu et al., 2013*; *Cho et al., 2014*). We tested the most likely frameshift insertion and deletion (indel) mutations that could be introduced by unexpected non-homologous end joining or inaccuracies in the sgRNA targeting. We used PCR amplification and sequencing to determine that there were no mismatches in these sequences compared to wildtype controls (*Appendix 1—table 2* and *Supplementary file 1*). We are thus confident that our LoxP insertion did not cause any genetic mutations that would interfere with in vivo studying of the GR functioning in the rats on a genetic basis. As an additional control, we further investigated physiological and behavioral effects of this targeted insertion. We performed multiple tests of behavior (open field, elevated plus maze, and FST), and observed no effect of gene targeting on any behavioral endpoint (*Appendix 1—figure 3*).

Furthermore, we found no differences in bodyweight (*Appendix 1—figure 4*) or organ weights commonly changed with chronic stress (*Appendix 1—table 3*).

Our functional studies indicate that female (but not male) rats with knockdown of GR in the PL-PFC have heightened fear response and impaired extinction. Importantly, these data inform prior PL-PFC lesion/inactivation studies, which indicate that the PL-PFC is critical for appropriate encoding and expression of conditioned fear (*Quirk et al., 2000*; *Morgan and LeDoux, 1995*; *Marquis et al., 2007*). The data indicate that GR signaling is an important component of this process in females, but may be less so in males (although the increased freeze times in males may reflect a more 'intense' response to the shock, which may result in ceiling effects on subsequent exposure to cues). In the FST, males but not females increased the number of diving events. As diving is considered an active coping behavior (*Bielajew et al., 2003*; *Fujisaki et al., 2003*; *Molina et al., 1994*), it is likely that GR in the PL-PFC also interfaces with coping behaviors in a sex-specific manner. In combination, these data indicate that use of enhanced precision methods of gene targeting such as CRISPR/Cas9 in rat reveal substantive new information on the biology of the PL-PFC and its interaction with biological sex.

Deficits in conditioned fear and glucocorticoid feedback efficacy are associated with depression, post-traumatic stress disorder and other stress related diseases in human. It is important to consider that these disease states are overrepresented in women (*McKlveen et al., 2013*; *Herman, 1993*), and also involve modification of prefrontal cortical circuitry in neuroimaging studies (*Breslau et al., 1997*; *Breslau and Anthony, 2007*; *Kessler et al., 1995*; *McLean et al., 2011*). Emergence of behavioral deficits following targeting of GR in the PL-PFC of females may reflect a mechanism underlying selective vulnerability of females to stressful life events, suggesting that appropriate GR signaling is required to mitigate the impact of adversity.

Overall, we demonstrate that CRISPR/Cas9 gene editing is effective in generating novel tools for bio-behavioral research in a highly tractable model organism with a long and well-documented history. As noted in our functional studies, the SD:nr3c1$^{fl/fl}$ line can be of significant value in the context of higher order behavioral assays (cognitive behaviors, goal-directed behaviors, reward behaviors/ drug self-administration) that can be often difficult to implement and/or interpret in other rodent organisms, such as mice. With the growing number of viral Cre constructs and the recent surge in development of rat Cre driver lines, the CRISPR/Cas9 method can provide site and cell type gene deletions or manipulations that are vital for our understanding of mechanisms of stress pathologies and stress-related diseases.

## Materials and methods

### Generation of the floxed *Nr3c1* line

Six sgRNAs targeting the sequences flanking exon 3 of *Nr3c1* were designed using the CRISPR Design Tool website (http://www.genome-engineering.org/). The complementary oligos (IDT) with overhangs were cloned into the BbsI site of the pX459 vector (Addgene #48139), according to the published methods (*Ran et al., 2013*). Editing activity was validated by the T7E1 assay (NEB) in rat C6 glioma cells (ATCC), compared side-by-side with *ApoE* sgRNA that was previously shown to work efficiently in rat zygotes (*Ma et al., 2014*). Validated sgRNA were in vitro transcribed by MEGAshorscript T7 kit and then purified by MegaClear kit (ThermoFisher). *Cas9* mRNA was in vitro transcribed by mMESSAGE mMACHINE T7 ULTRA kit (ThermoFisher), according to manufacturer's instructions. Two sgRNAs (50 ng/ul ech), *Cas9* mRNA (100 ng/ul), and the donor plasmid (100 ng/ul) were mixed in 0.1X TE buffer and injected into the cytoplasm of one-cell-stage embryos of Sprague Dawley genetic background rats via a piezo-driven cytoplasmic microinjection technique. Injected embryos were immediately transferred into the oviductal ampulla of pseudopregnant females. Live born pups were genotyped by PCR, enzyme digestion, and Sanger sequencing. Rats were bred and housed in a vivarium with a 12 hr light/dark cycle. All animal studies were approved by the Institutional Animal Care and Use Committees of the Cincinnati Children's Hospital Medical Center and University of Cincinnati.

## Breeding and genotyping

A female Sprague Dawley rat (#60) containing the floxed *Nr3c1* alleles was generated by the Transgenic Animal and Genome Editing Core Facility (*Figure 1A and B*). The founder rat that was heterozygous for the LoxP knock-in sequences was crossed with a wildtype Sprague Dawley male rat. F1 heterozygous offspring were bred to generate F2 and F3 offspring. We used F2 and F3 heterozygote animals to generate SD:nr3c1$^{fl/fl}$ and SD:nr3c1$^{wt}$ littermate controls for behavioral and molecular experiments. Our breeding scheme was designed to minimize inbreeding by avoiding sibling x sibling mating.

We used four different sets of primers (*Figure 1C and D*) to verify that the LoxP sequences were inserted in the 5' and 3' sides of exon 3. DNA was extracted from tail blood sample by using PureLink Genomic DNA Kits (Invitrogen Cat #: K1820-01, K1820-02, K1821-04). To discriminate the genotype of the rats, we performed PCR reactions with the primers that are listed in *Figure 1C and D*, by using FailSafe PCR 2X PreMix D (Epicenter Cat # FSP995D) and Dream Taq enzyme (Thermofisher Cat # EP0701) in a PTC-200 Peltier Thermal Cycler. All of the PCR products were analyzed with agarose gel electrophoresis and the images were captured by Axygen Gel Documentation System (*Figure 1E–G*).

For confirmation by sequencing, the expected bands from homozygote rats (SD:nr3c1$^{fl/fl}$) and wildtype littermate controls (SD:nr3c1$^{wt}$) were separated from the agarose gel and purified with Thermo Scientific GeneJET Gel Extraction Kit. The purified DNA was sent to Cincinnati Children's Hospital Medical Center (CCHMC) DNA Sequencing and Genotyping Core to sequence the 3' and 5' sites of exon 3 of the *Nr3c1* gene, using the P1-P2 and P3-P4 primers. The sequencing data clearly showed the inserted flox sequences in both 3' and 5' sides of *Nr3c1* in SD:nr3c1$^{fl/fl}$ rats. The flox sequences were not seen in the SD:nr3c1$^{wt}$ controls (*Figure 1G*).

## Stereotaxic surgery

Adult male and female SD:nr3c1$^{fl/fl}$ and SD:nr3c1$^{wt}$ rats (250-350 g) were singly housed on a 12 hr light/dark cycle in a temperature- and humidity-controlled housing facility at the University of Cincinnati. All experimental procedures were conducted in accordance with the National Institutes of Health Guidelines for the Care and Use of Animals and were approved by the University of Cincinnati Institutional Animal Care and Use Committee. Animals were deeply anaesthetized with 4–5% Isoflurane, prior to placement in the stereotaxic frame (Kopf Instruments) and sedation maintained at 2–3% isoflurane during surgery. A 2 ul Hamilton syringe was used to administer viral constructs. The needle was gently lowered to the predefined coordinates for BLA (AP: −2.7, ML:±4.8, DV: 8.8) or PL-PFC (AP:+3.0, ML:±0.6, DV:−3.3) and a 5 min rest period was observed. The virus was infused over 5 min (1 ul/5mins). After infusion the needle remained in place for an additional 5 min. The needle was slowly removed and the hole sealed with gelfoam. After completion of all infusions, the surgical site was closed with surgical staples and animals were singly housed for recovery and for the remainder of the studies.

## Viral constructs

AAV8-hSyn-Cre (titer: 6.5 × 10$^{12}$ molecules/ml) was sourced from the UNC Vector Core (Chapel Hill, NC, USA). AAVrg-CAG-GFP (titer: 5 × 10$^{12}$ vg/mL, this construct was a gift from Edward Boyden to Addgene- viral prep # 37825-AAVrg) and AAVrg pmSyn1-EBFP-Cre (titer: 6 × 10$^{12}$ vg/mL, this construct was a gift from Hongkui Zeng to Addgene -viral prep # 51507-AAVrg; *Madisen et al., 2015*), were sourced from Addgene (MA, USA). All constructs were administered 1 ul bilaterally and a minimum of 3 weeks incubation was allowed.

For CaMKIIα cell-specific knockdown and fear conditioning and forced swim studies, 0.1 ul of AAV9.CamKII.HI.eGFP-Cre.WPRE.SV40 (CaMKIIα Cre) [Penn Vector Core] (titer: 6.544 × 10$^{13}$ diluted to 6.544 × 10$^{11}$] was injected bilaterally and allowed 5 weeks to incubate.

## Immunohistochemistry

3–5 weeks after viral injection animals were injected with an overdose of pentobarbital and perfused transcardially with 0.1M PBS until blood was clear, followed by 4% paraformaldehyde/0.1M PBS for 15 min. The brains were postfixed overnight at 4°C in 4% paraformaldehyde/0.1M PBS, rinsed 2 × 3 times with 0.1M PBS and cryoprotected in 0.1M PBS containing 30% sucrose + 0.01% sodium azide

at 4°C until the brains sank in the sucrose solution. Brains were frozen to −20°C on the stage of a sliding microtome (Leica Biosystems Inc, Nußloch, Germany) and sectioned at 35 µm, collected in a series of 6 and placed in cryoprotectant consisting of 10% polyvinyl-pyrrolidone- Mol Wt 40,000, +500 ml of 0.1M PBS + 300 ml of ethylene glycol +30% sucrose and 0.01% sodium azide. A full series of 6 sections was labeled for GFP, GR and Cre Recombinase. For viral verification of AAV8-hSyn-Cre (*Figure 2A–B*), AAVrg pmSyn1-EBFP-Cre (*Figure 2E*), and AAVrg-CAG-GFP (*Figure 2F*), the sections were processed as follows: incubtaed with 1% sodium borohyhdride/0.1M PBS for 30 min, rinsed 6 × 5 min in 0.1M PBS, incubated with 1% hydrogen peroxide/0.1M PBS for10 min, rinsed 6 × 5 minin 0.1M PBS, rinsed additionally in 0.1M PBS 4 × 15 min. Sections were blocked by incubation 0.1M PBS containing 4% goat serum with 0.4% Triton X-100% and 0.2% BSA for 2 hr, followed by a cocktail including polyclonal rabbit GR antibody (Santa Cruz-Cat# sc 1004) diluted 1:500, monoclonal mouse Cre Recombinase antibody (Millipore Cat # MAB 3120) diluted 1:1000 and no GFP antibody label (native virus expression) in blocking solution consisting of 0.1M PBS containing 4% goat serum with 0.4% Triton X-100% and 0.2% BSA overnight at room temperature. After overnight antibody incubation, sections were rinsed in 0.1M PBS 3 × 5 min, then incubated in a cocktail of goat-anti-mouse CY3 conjugated IgG (Invitrogen-Cat#A32727) diluted 1:800 and goat anti rabbit CY5 conjugated IgG (Invitrogen-Cat # A32733) diluted 1:800 in 0.1M PBS for 45 min, and rinsed 4 × 5 min in 0.1M PBS. All sections were mounted and viewed on a Nikon C2 Plus Confocal Microscope.

For AAV9.CamKII.HI.eGFP-Cre.WPRE.SV40 (CaMKIIα Cre) (*Figure 2C–2D*) cell-specific knockdown, brains were processed the same until immunohistochemistry. The sections were processed as followed: slices rinsed 5 × 5 in 0.1M PBS, incubation in 1.5% 10 mM Trisodium Citrate in PBS at 80°C for 10 min, rinsed 5 × 5 00.1M PBS, blocked in 0.2% BSA +4% goat serum 1 hr room temperature, and then incubated overnight at 4°C in a cocktail of block plus rabbit polyclonal GR antibody (Invitrogen-Cat# PA1-511A) diluted 1:800, chicken polyclonal GFP antibody (abcam-Cat# ab13970) diluted 1:2000, and mouse monoclonal CaMKII antibody (abcam-Cat# ab22609) diluted 1:250. The following day sections were washed 5 × 5 in 0.1M PBS and incubated at room temperature for 1 hr in the following secondary antibodies diluted 1:500 in blocking solution: goat-anti-Rabbit Cy3 conjugated IgG (Invitrogen-Cat# A10520), goat-anti-chicken conjugated Alexa 488 (Invitrogen-Cat# A11039), and goat-anti-mouse CY5 conjugated IgG (Invitrogen-Cat# A10524).

Verification of GR knockdown for functional PL-PFC knockdown study (AAV9.CamKII.HI.eGFP-Cre.WPRE.SV40) was completed via immunohistochemical analysis of GR protein (*Figure 3C–D*). Sections (35 µm, series of 6) were processed as stated above before immunohistochemistry. The sections were processed as follows: incubated with 1% sodium borohydride/0.1M PBS for 30 min, rinsed 6 × 5 min in 0.1M PBS, incubated with 1% hydrogen peroxide/0.1M PBS for 10 min, rinsed 6 × 5 min in 0.1M PBS, rinsed additionally in 0.1M PBS 4 × 15 min. Sections were blocked by incubation with 0.1M PBS containing 4% goat serum with 0.4% Triton X-100% and 0.2% BSA for 2 hr, followed by incubation with rabbit polyclonal GR antibody (Invitrogen-Cat# PA-521341) diluted 1:500 [and no GFP antibody label (native virus expression)] in blocking solution overnight at room temperature. After overnight antibody incubation, sections were rinsed in 0.1M PBS 3 × 5 min, then incubated in a cocktail of goat-anti-mouse CY3 conjugated IgG (Invitrogen-Cat#A32727) diluted 1:800 in 0.1M PBS for 45 min, and rinsed 4 × 5 min in 0.1M PBS. All sections were mounted and viewed on a Nikon C2 Plus Confocal Microscope.

The validation of the GR knockdown was performed by obtaining relative intensity of GR expressing cells in both control and knockdown rats within and outside viral infected cells. This was performed in a semi quantitative manner (non-calibrated to known GR concentration). We used a partially automated analysis in which the software (ImageJ software -U. S. National Institutes of Health, Bethesda, Maryland, USA) defines a level of background intensity in which GR/GFP is considered positive, and positive nuclei are outlined based on circularity and size which were previously manually validated. Specifically, the total number of GFP expressing cells (virus infected cells) in the PL-PFC was determined on the green channel. Using the outline of the GFP cells, a region of interest (GFP-ROI) was created (see representative image, *Figure 3*). On the red channel (GR), the total number of GR expressing cells was selected and the mean optical density (OD) was obtained (Total GR OD). This reflects the total intensity of GR staining in each image. Subsequently, the GFP-ROI was overlaid on to the red channel image to obtain the uncalibrated OD inside the GFP-ROI, corresponding to GR expression within virus infected cells. Note that subtraction of background optical density was not performed.

Verification of virus placement for the functional PL-PFC GR knockdown study (AAV9.CamKII.HI.eGFP-Cre.WPRE.SV40) was completed by immunohistochemical analysis of Cre recombinase protein (*Figure 3E*). Sections (series of 6) were processed as stated above before immunohistochemistry. Sections were washed 5 × 5 in 0.1M PBS and blocked in 0.2%BSA +0.4% Triton-X 100 + 4% goat serum. Sections were then incubated overnight in block at 4°C in monoclonal mouse cre recombinase antibody (Millipore-Cat# MAB 3120) diluted 1:1000. The following day sections were washed 5 × 5 in 0.1M PBS and incubated in goat-anti-mouse CY3 conjugated IgG (Invitrogen A32727) diluted 1:500 at room temperature for 1 hr. All sections were mounted and viewed on a Nikon C2 Plus Confocal Microscope.

## Auditory fear conditioning

We used the (*Quirk et al., 2000*) method for auditory fear conditioning. Rats were placed into sound attenuated chambers (Med Associates Fairfax VT) with a 33×28×25 cm interior chamber within a 63×45×58 cm sound attenuating outer box and aluminum walls and aluminum rod floor. The fear conditioning paradigm was controlled by Ethovision software (Noldus Information Technology) and consisted of 3 days. On day one, (acquisition) rats were allowed to habituate to the chamber for 5 min followed by 5 x 30 s tones paired with shock, 0.5mA for 0.5 s, administered with 3 min inter-trial intervals (ITI). Rats were removed to their homecage and returned to the chamber 24 hr later. For day two (extinction), after another 5 min habituation, 20 x 30 s tones, with 3 min ITIs, were played with no shock at termination. Rats were again returned to the homecage for 24 hr. The last day, day three (extinction recall), consisted again of a 5 min habituation and 20 x 30 s tones, with 3 min ITIs. Freezing, the complete cessation of all movement other than respiration was measured by Freezescan software (CleverSys, Inc) during the 30 s tones.

## Forced swim test (FST)

The FST was conducted in regular lighting 1–4 hr after the beginning of the light cycle. On day 1, rats were placed in a Plexiglas cylinder measuring 61 cm deep with a 19 cm diameter filled to 40 cm with tap water at 24–26°C for 10 min. On day 2, rats were again placed in the apparatus under the same conditions for 10 min. Sessions were video-recorded and behavior was later analyzed by recording whether the rat was swimming, climbing, diving, or immobile using Kinoscope software (*Kokras et al., 2017*) by an experimenter blind to genotype.

## Corticosterone measurements after acute restraint

Rats were restrained in a well ventilated plastic restrainer for 30 min. Blood samples were collected in EDTA containing microtubes via tail nick (well below the last vertebrae) with a sterile razor blade. Blood samples were taken at the following time points: 0 min – basal before restraint, 15 min and 30 min while in the restrainer, and 60 min and 120 min after being released back into the home cage. Blood was kept on ice through the restraint and then centrifuged to remove plasma. Plasma was kept at −20°C until processed with a I-125 radioimmunoassay (RIA) kit from MP Biomedicals. Duplicates were run for each sample for technical replication.

## Statistical analysis

Data are expressed as mean ± standard error of the mean (SEM). An unpaired t-test was used to analyze GR knockdown in females and males. A two-way repeated measures analysis of variance (ANOVA) was used to analyze fear conditioning data (parametric) and a general linear model was used for analysis of corticosterone data (non-parametric). Bonferroni was used for post hoc analyses when interactions were significant and also for a priori planned comparisons between genotypes. No data points were excluded as outliers. There were eight data points excluded on female fear conditioning for extinction recall due to software technical error. The forced swim testFST was analyzed using Student's two-tailed T-test. Behavioral data was scored by a researcher blinded to GRKD condition. Sigmaplot 13.0 (Systat Software) was used to analyze the data. GraphPad Prism 8 (San Diego, CA) and Sigmaplot 13.0 (Systat Software) was used to graphical present the data. For behavioral experiments, sample size was dependent on the outcome of breeding. Target 'n' was 10/group, based on previous power analyses performed in our group; however, 'n's were decreased due to missed or ineffective viral injections in the GRKD groups. Effect sizes were calculated to

assess the strength of our findings in the face of reduced 'n's'. Effect sizes were in the in the small to medium (pη2, ANOVA) and medium to large (Cohen's d, t-tests) range.

## Acknowledgements

J Herman is supported by NIH grants (R01 MH049698, R01 MH101729). R Moloney is supported by a NARSAD Young Investigator Award from the Brain and Behavior Research Foundation. J Scheimann is supported by University of Cincinnati University Research Council Student Faculty Collaboration grant. Y Hu is supported by the Transgenic Core Director Endowment Fund from Cincinnati Children's Research Foundation. E Cotella is supported by an NIH grant (T32 DK059803). We would also like to thank members of the Herman lab for technical assistance and the Transgenic Animal and Genome Editing Core at Cincinnati Children's Hospital Medical Center for assistance in generating the floxed *Nr3c1* rat.

## Additional information

### Funding

| Funder | Grant reference number | Author |
|---|---|---|
| National Institute of Mental Health | R01 MH049698 | James P Herman |
| National Institute of Mental Health | R01 MH101729 | James P Herman |
| Brain and Behavior Research Foundation | NARSAD Young Investigator | Rachel D Moloney |
| University of Cincinnati | University Research Council Student Faculty Collaboration grant | Jessie R Scheimann |
| National Institute of Mental Health | T32 DK059803 | Evelin M Cotella |
| Cincinnati Children's Research Foundation | | Yueh-Chiang Hu |

The funders had no role in study design, data collection and interpretation, or the decision to submit the work for publication.

### Author contributions

Jessie R Scheimann, Rachel D Moloney, Conceptualization, Data curation, Formal analysis, Validation, Investigation, Visualization, Methodology, Writing—original draft, Writing—review and editing; Parinaz Mahbod, Conceptualization, Formal analysis, Validation, Visualization, Methodology, Writing—original draft, Writing—review and editing; Rachel L Morano, Formal analysis, Validation, Investigation, Visualization, Methodology, Writing—review and editing; Maureen Fitzgerald, Validation, Investigation, Visualization, Methodology, Writing—review and editing; Olivia Hoskins, Formal analysis, Investigation, Writing—review and editing; Benjamin A Packard, Conceptualization, Validation, Investigation, Methodology, Writing—review and editing; Evelin M Cotella, Software, Formal analysis, Investigation, Visualization, Methodology, Writing—review and editing; Yueh-Chiang Hu, Conceptualization, Resources, Data curation, Software, Formal analysis, Validation, Investigation, Visualization, Methodology, Writing—original draft, Project administration, Writing—review and editing; James P Herman, Conceptualization, Resources, Software, Supervision, Funding acquisition, Methodology, Writing—original draft, Project administration, Writing—review and editing

### Author ORCIDs

Jessie R Scheimann https://orcid.org/0000-0001-7912-4898
Rachel D Moloney https://orcid.org/0000-0001-7111-3414

### Ethics

Animal experimentation: All experimental procedures were conducted in accordance with the National Institutes of Health Guidelines for the Care and Use of Animals and were approved by the University of Cincinnati Institutional Animal Care and Use Committee (#04-08-03-01).

### Decision letter and Author response

Decision letter https://doi.org/10.7554/eLife.44672.017
Author response https://doi.org/10.7554/eLife.44672.018

## Additional files

### Supplementary files

• Supplementary file 1. List of primers used to identify the off-target events.
DOI: https://doi.org/10.7554/eLife.44672.006

• Transparent reporting form
DOI: https://doi.org/10.7554/eLife.44672.007

### Data availability

All data generated or analyzed during this study are included in the manuscript and supporting files.

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

# Appendix 1

DOI: https://doi.org/10.7554/eLife.44672.008

## Supplementary information

The individual editing activity was validated in rat C6 glioma cell line by T7E1 assay, compared side-by-side with a control sgRNA targeting rat *ApoE*. Cells were transfected with individual sgRNAs (*Appendix 1—table 1* lists the sgRNAs tested) and cultured for two days. Cells were then harvested for DNA extraction and T7E1 assay. The editing activity was calculated as percentage of the cut band intensity over the total band intensity. The data were represented as relative fold change respective to the control (*Appendix 1—figure 1*).

**Appendix 1—table 1.** List of sgRNA chosen to target exon 3 of the *Nr3c1* gene (glucocorticoid receptor) for insertion of LoxP sites.

| Target location | ID | sgRNA target sequence |
|---|---|---|
| 5' to exon 3 | sg-1 | TAAGGTGAACAGTAAACTAC |
| | sg-2 | GGAAGGGAAAGGTCTAT |
| | sg-3 | GTCATTTGATAGCATGGCA |
| 3' to exon 3 | sg-4 | GAGCAGCTTTATTTTGGA |
| | sg-5 | GATTTGCTGTGAGATCAATC |
| | sg-6 | GATCCAAAGGAAACTGT |
| control | ApoE sgRNA | GGTAATCCCAGAAGCGGTTC |

DOI: https://doi.org/10.7554/eLife.44672.009

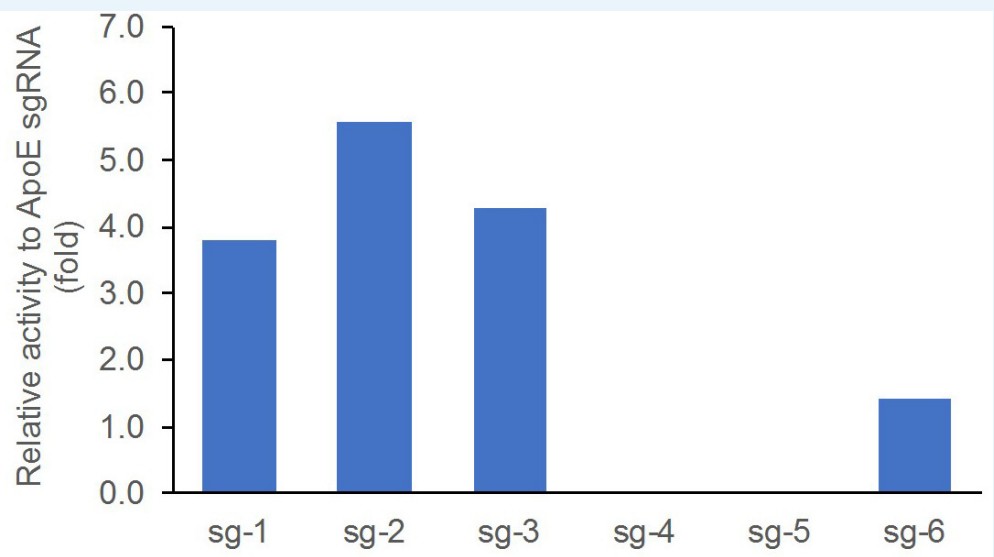

**Appendix 1—figure 1.** Validation of sgRNA editing activity by the T7E1 assay in rat C6 glioma cells, compared side-by-side with *ApoE* sgRNA that was previously shown to work efficiently in rat zygotes. Data presented as relative fold change between the percentage of cut band intensity over total band intensity induced by the individual sgRNAs and that of the control ApoE sgRNA.

DOI: https://doi.org/10.7554/eLife.44672.010

## Off-target sequences

Off-target sequences associated with each sgRNA that were used to design the SD:nr3c1[fl/fl] rat were carefully considered (**Appendix 1—table 2**). The required primers (**Supplementary file 1**) were designed with Primer3 software Web for the sequences with highest off-target score, based on the CRISPOR Webtool (http://crispor.tefor.net). PCR products were purified with Thermo Scientific GeneJET Gel Extraction Kit and sequenced by the DNA Sequencing Core at Cincinnati Children's Hospital Medical Center. We performed off-target analysis on the generated chromatograms by comparing peak-to-peak and nucleotide-to-nucleotide signatures between SD:nr3c1[wt] and SD:nr3c1[fl/fl] rats. We observed no off-target events among the most likely off-target sites (**Appendix 1—table 2**).

**Appendix 1—table 2.** List of off-target sequences used to identify the off-target events.

**Off-target sequences**

| OT | 5' guide sequence | off-target sequence | Location | Strand | Locus description | Mismatch found |
|---|---|---|---|---|---|---|
| OT-1 | TCTGGAAGGGAA AGGTCTAT AGG | TTGAGAACGGA AAGGTCTAT GGG | chr19:18822843– 18822865 | - | intergenic: AABR07043040.1- AABR07043071.1 | 0 |
| OT-2 | TCTGGAAGGGAA AGGTCTAT AGG | TTGAGAACGGA AAGGTCTAT GGG | chr19:18815538– 18815560 | - | intergenic: AABR07043039.1- AABR07043040.1 | 0 |
| OT-3 | TCTGGAAGGGAA AGGTCTAT AGG | GTAAGAAGGGA AAGGTCTAT GGG | chr7:13815358– 13815380 | + | intergenic: Slc1a6-Cyp4f37 | 0 |
| OT-4 | TCTGGAAGGGAA AGGTCTAT AGG | TATGCAGAGGA AAGGTCTAT TGG | chr3:171628668– 171628690 | + | intergenic: LOC689618- Rab22a | 0 |
| OT-5 | TCTGGAAGGGAA AGGTCTAT AGG | TAAGCAAGTGAA AGGTCTAT AGG | chr4:25084248– 25084270 | + | intergenic: U1-Steap1 | 0 |
| OT-6 | TCTGGAAGGGAA AGGTCTAT AGG | GCAGTAAGGGG AAGGTCTAT GGG | chr6:38673626– 38673648 | + | intron:Nbas | 0 |
| OT-7 | TCTGGAAGGGAA AGGTCTAT AGG | ACAGGAAGGAT AAGGTCTAT TGG | chr15:91803840– 91803862 | + | intergenic: AABR07019155.1- AABR07019162.1 | 0 |
| | 3' guide sequence | off-target sequence | Location | strand | locus description | mismatch found |
| OT-8 | AGAGATCCAAAG GAAACTGT GGG | AATTCTCCAAAG GAAACTGT GGG | chr14: 46180580– 46180602 | - | intergenic:Nwd2- AABR07015040.1 | 0 |
| OT-9 | AGAGATCCAAAG GAAACTGT GGG | GGAAATGCACAG GAAACTGT GGG | chr13:47968158– 47968180 | - | intron:Rassf5 | 0 |
| OT-10 | AGAGATCCAAAG GAAACTGT GGG | ACACAGCGAAAG GAAACTGT GGG | chr7:101131361– 101131383 | - | intron: LOC500877 | 0 |
| OT-11 | AGAGATCCAAAG GAAACTGT GGG | ATGAATCCTAAGG AAACTGT AGG | chr2:189985378– 189985400 | - | intergenic: U1/S100a3- S100a3 | 0 |
| OT-12 | AGAGATCCAAAG GAAACTGT GGG | TGAGCTCCTGAG GAAACTGT AGG | chr11:75363850– 75363872 | + | intergenic: Hrasls-Mb21d2 | 0 |

DOI: https://doi.org/10.7554/eLife.44672.011

## Copy number analysis

DNA was extracted from brain samples of two SD:nr3c1[wt] (WT) and two SD:nr3c1[fl/fl] (f/f) rats by Purelink genomic DNA extraction kit (ThermoFisher, Cat# K1820-01). qPCR was performed in triplicates using primers ordered from IDT and Power SYBR Green PCR Master Mix (ThermoFisher, Cat#: 4367659) with QuantStudio 7 Pro Real-Time PCR Systems. 18 ng/ul gDNA from each sample was used for each reaction. Following the quantifying cycles, the

melt curve analysis was performed for each reaction to make sure there was a single product. The copy number was determined by the double delta Ct analysis. The delta Ct value from each region of the WT samples was set as two copies per genome (*Appendix 1—figure 2*).

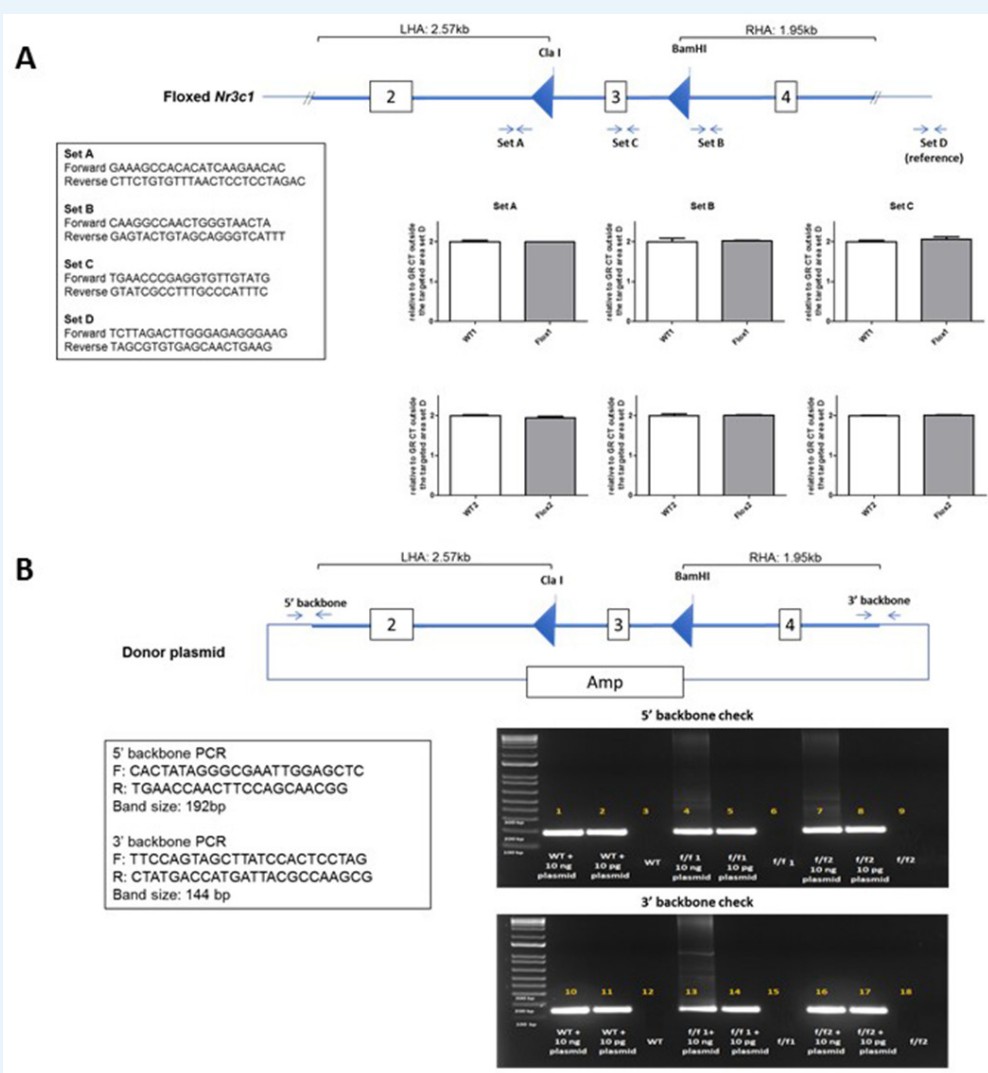

**Appendix 1—figure 2.** Screen for random integration of the donor plasmid in homozygous *Nr3c1* floxed rats. (**A**) SYBR Green-based qPCR was used to determine the copy number of the transgene in rats. Three regions of the integrated area were chosen for the analysis (Set A, Set B, and Set C). A region outside of the integrated area was used as an internal reference (Set D). Data presented as relative copy number of each region between the SD: nr3c1$^{wt}$ (WT) and SD:nr3c1$^{fl/fl}$ (f/f) samples. The delta Ct value of the WT sample as set as two copies per genome. (**B**) PCR amplifying two distant regions of the donor plasmid backbone was performed to detect the random integration event. A limited amount of the donor plasmid DNA was spiked into the WT genomic DNA as positive control.

DOI: https://doi.org/10.7554/eLife.44672.012

## Behavioral characterization

GR floxed Heterozygotes from the NF2 generation were bred together producing three genotypes, SD:nr3c1$^{fl/fl}$ (f/f), SD:nr3c1$^{fl/-}$ (het), and SD:nr3c1$^{wt}$ (wt). Only f/f and wt littermate controls were used for behavioral and physiological measures. All litters were born within 4 days of one another and were 8 weeks of age at the beginning of behavioral assays. Rats were single housed during behavioral experiments in a temperature and humidity controlled

vivarium on a 12/12 hr light cycle with ad libitum access to food and water. (n = 12 female f/f and n = 12 female wt; n = 12 male f/f and n=12 male wt).

Elevated plus maze (EPM) and Open field (OF) tests were performed in dim light 1–4 hr after the beginning of the light cycle. For the EPM, rats were placed on an elevated plus maze with two closed arms ipsilateral from one another and two ipsilateral open arms. Arms measured 10 cm x 50 cm and were elevated 62 cm from the ground. Rats were allowed to freely explore the maze for 5 min where time spent in open arms and closed arms was analyzed by Ethovision software (Noldus Information Technology, Wageningen - The Netherlands). For the OF, rats were placed in an uncovered plastic square arena measuring 92 cm × 92 cm and allowed to explore freely for 5 min. The number of entries into a square center area measuring 45 cm x 45 cm was analyzed, along with total locomotion, by Ethovision software (Noldus Information Technology, Wageningen - The Netherlands). Animals also underwent the forced swim test (FST) as described in the main text.

The EPM is a common test for anxiety like behavior. Time spent exploring open arms is considered as less anxious, while more anxious animals tend to stay in closed arms where there are less environmental threats. Male and female rats from f/f or wt groups all spend equal amounts of time exploring open arms [Male $T(20) = -1.207$; p=0.241 (*Appendix 1—figure 3A*); Female $T(19) = -0.0939$; p=0.926 (*Appendix 1—figure 3B*)].

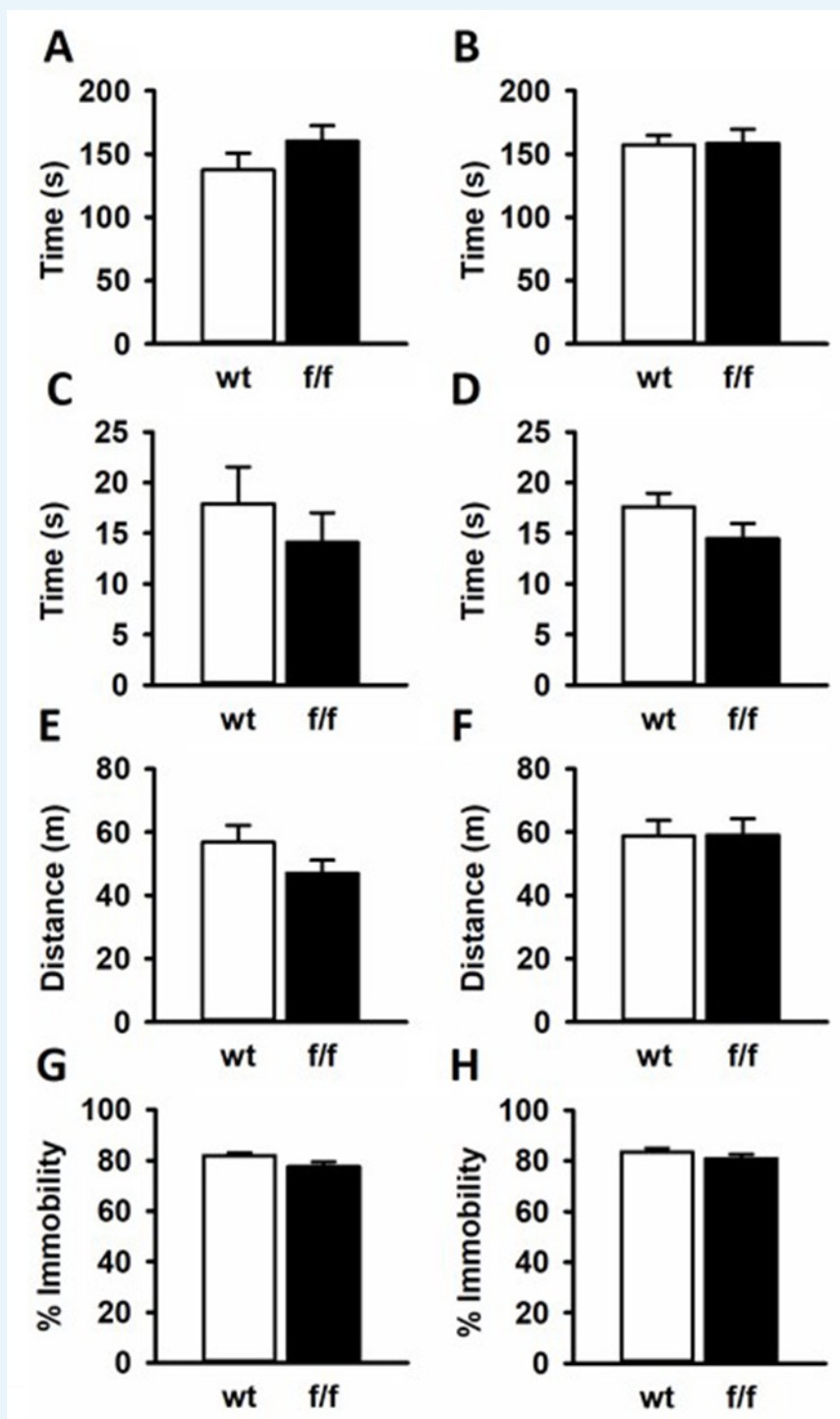

**Appendix 1—figure 3.** SD:nr3c1$^{fl/fl}$ (f/f) rats did not show behavioral differences from SD: nr3c1$^{wt}$ (wt) controls in common behavioral assays. Male (**A**) and female (**B**) f/f and wt controls spent equal time in the open arms of the elevated plus maze. Males (**C**) and females (**D**) showed no differences in time spent in the center of the open field. No locomotor phenotype was observed in male (**E**) and female (**F**) f/f rats compared to wt controls in the open field. There were no differences in immobility in the forced swim test for male (**G**) or female (**H**) f/f rats compared to controls.

DOI: https://doi.org/10.7554/eLife.44672.013

Similarly, the OF is a test for anxious behavior with anxious rats spending the majority of the 5 min OF test around the periphery where the rodent feels less exposed to environmental dangers. Less anxious rats will explore the center of the arena more. Any alterations to locomotion in animal models can also be seen in the OF test. When tested in the OF, GR floxed rats explored the center for the same amount of time as wt controls in both males [T(20)=0.762; p=0.455 (*Appendix 1—figure 3C*] and females [T(19)=−1.563; p=0.134 (*Appendix 1—figure 3D*)]. The f/f male rats and wt male rats showed no differences in locomotion in the maze [T(20)=1.365; p=0.187 (*Appendix 1—figure 3E*)], and the same was true for the female f/f and female wt rats [T(19)=−0.0326; p = 0.974 *Appendix 1—figure 3F*].

In studies focusing on the effects of stress, the FST is often used as an assay to test for the efficacy of antidepressants to attenuate phenotypes (*Cryan et al., 2005*). Antidepressants have been shown to decrease immobility, considered passive coping behavior. Further the FST tests for the rodents coping mechanism and how that can change after a manipulation such as chronic stress (*McKlveen et al., 2013*; *Ghosal et al., 2014*). There were no differences in FST immobility for male [T(20)=1.960; p=0.0641 (*Appendix 1—figure 3G*)] nor female f/f and wt rats [T(19)=1.178; p=0.253 (*Appendix 1—figure 3H*)].

## Physiology measures

To verify that the gene editing protocol had no effect on peripheral physiology, animals were weighed either weekly or bi-weekly from weaning, and heart, thymus, and adrenals were dissected from animals after euthanasia (rapid decapitation) and weighed. No differences were observed in body weight profiles in either male (*Appendix 1—figure 4A*) or female (*Appendix 1—figure 4B*) SD:nr3c1$^{fl/fl}$ relative to SD:nr3c1$^{wt}$ controls. Similarly, weight of peripheral organs (thymus, adrenals, heart) were not affected by gene targeting (*Appendix 1—table 3*).

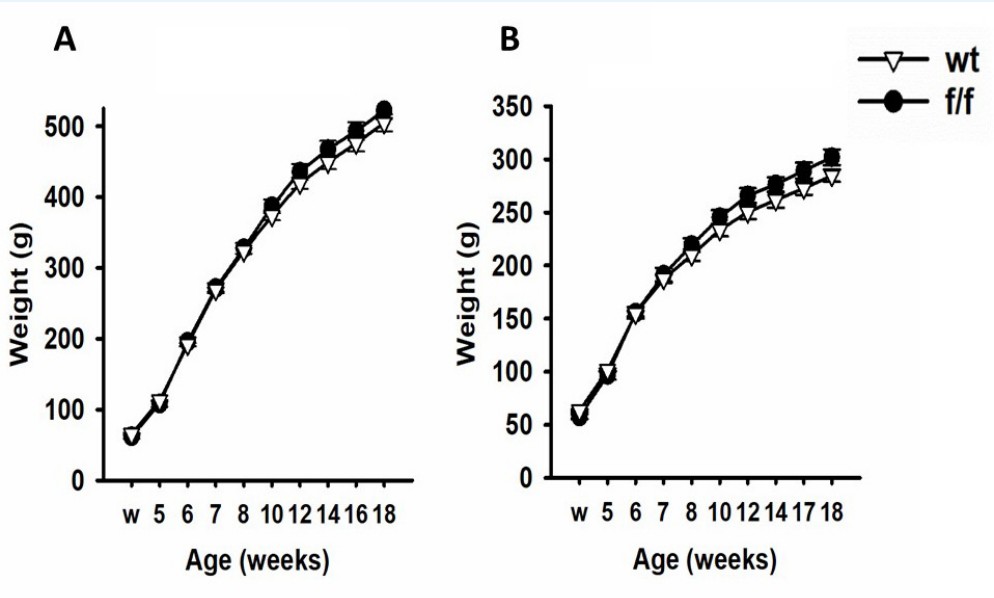

**Appendix 1—figure 4.** SD:nr3c1$^{fl/fl}$ (f/f) rats did not differ in bodyweight compared to SD: nr3c1$^{wt}$ (wt) controls. Male (**A**) and female (**B**) fl/fl and wt rats did not differ in weight at weening (w) or over the course of the experiment.
DOI: https://doi.org/10.7554/eLife.44672.014

**Appendix 1—table 3.** Hearts, thymi, and adrenal weights did not differ in fl/fl vs. wt controls.

| Males | Organ | Wt weight (mg) | St error | F/f Weight (g) | St error | T-test | P value |
|---|---|---|---|---|---|---|---|
| | Heart | 1284.762 | 37.318 | 1272.990 | 21.168 | $T_{(20)} = 0.286$ | p=0.778 |
| | Thymus | 302.546 | 15.669 | 272.880 | 16.244 | $T_{(20)} = 1.299$ | p=0.209 |
| | Adrenals Averaged | 28.188 | 0.565 | 29.050 | 1.266 | $T_{(20)} = -0.581$ | p=0.568 |
| Females | Organ | wt Weight (mg) | St Error | f/f Weight (g) | St Error | T-test | P value |
| | Heart | 805.408 | 60.710 | 835.467 | 16.912 | $T_{(19)} = -0.417$ | p=0.681 |
| | Thymus | 211.383 | 13.307 | 230.733 | 9.894 | $T_{(19)} = -1.097$ | p=0.287 |
| | Adrenals Averaged | 45.100 | 2.113 | 42.633 | 1.007 | $T_{(19)} = 0.947$ | p=0.355 |

DOI: https://doi.org/10.7554/eLife.44672.015

## Statistical analysis

Data are expressed as mean ± standard error of the mean (SEM). Student's two-tailed T-test was used to analyze behavioral data (EPM, OF and FST) and organ weights. Body weight was analyzed by two-way repeated measures ANOVA. No data points were excluded as outliers. Behavioral data was scored by a researcher blinded to condition on Ethovision software (Noldus Information Technology, Wageningen - The Netherlands). Sigmaplot 13.0 (Systat Software) was used to analyze the data and for graphical representation of the data. For behavioral experiments, sample size was dependent on the outcome of breeding. Target 'n' was 10/group, based on previous power analyses performed in our group.

