## [Decision Letter]

Thank you for submitting your article "Use of CRISPR/Cas9 gene targeting to conditionally delete glucocorticoid receptors in rat brain" for consideration by *eLife*. Your article has been reviewed by three peer reviewers, and the evaluation has been overseen by Geoffrey Schoenbaum as the Reviewing Editor and Catherine Dulac as the Senior Editor. The following individuals involved in review of your submission have also agreed to reveal their identity: Brandon K Harvey (Reviewer #1) and Carmen Sandi (Reviewer #3).

The reviewers have discussed the reviews with one another and the Reviewing Editor has drafted this decision to help you prepare a revised submission.

Summary:

In the current study, the authors report the development of a conditional deletion of the GR in rats. They provide credible evidence that their tool works, affects GR conditionally and that this impacts behavior as expected. The reviewers agreed that having such a tool available in rats would be a substantial step forward for the field. There are concerns but they are requests for details or changes in emphasis, none of which should take more than a few months. If the paper is revised with the additional details, the reviewers agreed that it would be acceptable for publication.

Essential revisions:

The essential revisions are targeted at providing additional detail that was thought to be critical to the validity of the tool. Other details requested in the reviews are important, but these are particularly critical as they support the rationale for publishing. There are three:

1) Correcting the emphasis on CRISPR as raised by reviewer 1.

2) Immunostaining patterns comparing effects in males and females as raised by reviewer 2.

3) Sequencing of PCR amplicons for the entire integrated transgene as raised by reviewer 1.

*Reviewer #1:*

The authors have developed a conditional knockout rat based on the Cre-loxP system to remove exon 3 of the Nr3c1 gene encoding the glucocorticoid receptor. They employ CRISPR/Cas9 in the development of a conditional knockout rat for the Nr3c1 gene. There is an overemphasis on the use of CRISPR/Cas9 starting with the title which is misleading in its current form. The conditional deletion of GR in rat brain is conditional on Cre not Cas9. Additionally, there is no evidence that CRISPR was required to make the animal. They obtained 1 animal out of 60 with the transgene. This could occur by injecting the targeting plasmid (as was done prior to the now common use of CRISPR for increased targeting). It is likely CRISPR aided in the process but there is no evidence for this (e.g. a plasmid alone was not injected and for good reason). The authors also do no appear to have done any PCR/Sequence/T7 assay/TIDE analysis on any of the 59 negative animals for evidence of CRISPR-induced mutation (which may have provided some possibly interesting GR mutants but this was not the goal). CRISPR should be deemphasized and focus on the fact there is a new Cre-dependent KO for GR and some possibly interesting biological observations using this new resource.

The authors claim in the first paragraph of the Results section that offspring fecundity is evidence of no random integration. This is not the case. There may be random integration that doesn't affect breeding but disrupts other gene function. A copy number per genome determination is needed. A PCR assay internal to the transgene that uses one of the primers partially overlapping the loxP site to avoid amplification of endogenous locus and a PCR assay to a single copy gene in the genome (2 copies per genome) should be used for QPCR to determine transgene copy number per genome. This assay is becoming a standard practice in transgenic animal development. Also, if the entire plasmid was injected, PCR assays designed to detect other plasmid segments is recommended.

The authors do not appear to have sequenced PCR amplicons for the entire integrated transgene. This is important to show no mutations or aberrant repair occurred at the single nucleotide level. Years of work may be built on these animals that have a point mutation in exon 3 altering function of GR. Alignment of sequencing data to in silico predicted sequence should be provided as supplement.

The use pf AAVrg-CAG-GFP as control for AAVrg pmSyn1-EBFP-Cre is not clear. These are different promoters and transgenes (e.g. using a "dead" Cre in the same construct would be preferred). Consistent with other experiments, AAVrg pmSyn1-EBFP-Cre should be injected into WT animals as control.

Overall, the finding of GR knockout in the prelimbic area affecting behavior is mostly confirmatory with published work while providing some interesting phenomenological observations for future studies. As such, this is a methods paper describing a transgenic rat with much potential and may be better in a specialized journal or methods section of neuroscience journal.

*Reviewer #2:*

Scheimann et al. generated a conditional GR knockout in rats using CRISPR/Cas9 technology. They provide evidence of targeted GR deletion in specific cell types, brain regions, and circuits via viral vector-mediated delivery of Cre. They also report that deletion of GR in output neurons of the prelimbic cortex resulted in sex-specific deficits in extinction of conditioned fear in females and a shift to active coping in the forced swim test in males. The manuscript is well-written and development of a conditional GR knockout model in rat is important and will be useful tool for probing GR function in the brain. However, there are several major limitations to the data that should be addressed to strengthen the conclusions of the manuscript.

1) Given the sex-specific effects on behavior, it is critical to show and evaluate the immunostaining patterns from both male and female rats. Are sex-specific differences observed in the distribution, efficiency, and/or extent of GR knockout in male and female brains?

2) Does the loss of GR lead to compensatory increases in the expression of the related receptor MR? This is an important consideration given that MR binds corticosterone, is co-expressed with GR in many brain regions, and is known to play a role in stress and emotional behavior.

3) Several issues should be addressed concerning Figure 2:

a) Why is the immunostaining pattern for GR in panel C different than in panel A? For the GR knockout cells in panel C, it appears that GR immunostaining is lost in the nucleus but not the cytoplasm. Why is this the case?

b) The authors use AAVrg-CAG-GFP injection of SD:nr3c1fl/fl rats as a control for the AAVrg pmSyn1-EBFP-Cre injection (panels E and F). Another important control is injection of AAVrg pmSyn1-EBFP-Cre into wild-type rats. Was this control performed?

c) The images should be enlarged. The small size of the images makes it very difficult to evaluate the staining patterns.

4) Does knockout of GR in the prelimbic cortex alter locomotor activity of the rats?

5) The sample size for the GR knockout rats is quite small (n = 5 for females and n = 6 for males) for behavioral studies. In addition, a Fisher's LSD test was employed for post-hoc analyses. Are similar significance results found when statistical tests are utilized that correct for multiple comparisons?

*Reviewer #3:*

This is the first study that develops a rat model with the capacity to perform cell- and region-specific deletion of the glucocorticoid receptor (GR) in rats. In addition to the novel tool development, the authors provided evidence of the behavioral impact of specific deletion of GR in the rat prelimbic CamKII-(glutamatergic) containing neurons. The data indicates differential effects of the knockdown approach on fear conditioning and extinction and behavior in the forced swim test in males and females, with a minor effect on corticosterone recovery in females. The study seems to be well performed and it is clearly written. Rats as model organisms have proved superior to mice for the dissection of stress (neuro)physiology and behavior and the current resource is a major step forward for the interrogation of GR function in this animal species.

[Editors' note: further revisions were requested prior to acceptance, as described below.]

Thank you for resubmitting your work entitled "Conditional deletion of glucocorticoid receptors in Rat brain results in sex specific deficits in fear&coping behaviors" for further consideration at *eLife*. Your revised article has been favorably evaluated by Catherine Dulac (Senior Editor), a Reviewing Editor, and two reviewers.

The manuscript has been improved but there are some remaining issues that need to be addressed before acceptance, as outlined below:

Please remove the statement concerning reviewer 1 below, and make certain the second issue raised is clearly noted as a caveat of the study if more data cannot be provided to address this directly.

*Reviewer #1:*

Below in quotes are the authors' responses to previous concerns. A follow up response is given. All other concerns were adequately addressed.

Authors: "…addition, we would like to clarify that we described the fact "the offspring of rat #60 can be bred to homozygosity indicates the correct targeting of loxP sequences to the Nr3c1 gene, instead of random integration". We did not mention anywhere in the text that offspring fecundity is evidence of no random integration."

My response: Although it is true the authors don't use the same verbiage, the ability to produce homozygous offspring should not be used as an indicator of correct targeting. The integration could still have additional integrants which would likely would be outcrossed if on a different chromosome or at least a sufficient distance from the targeted allele being genotyped. However, the only way to verify is to assay for such events and the authors have now done a more rigorous analysis to characterize the transgene integration. The statement in quotes provides an unnecessary precedent and contributes to the propagation of inadequate characterization of novel transgenic animals. It is not needed especially with the additional data and should be revised as simple statement of fact without indication or just remove it.

Authors: "The authors acknowledge that the use of these 2 viruses may not be entirely clear. The viruses used for these experiments were sourced from Addgene. At the time these experiments were performed, this combination of "control" and "Cre" viruses were the best available. The AAVrg pmSyn1-EBFP-Cre virus was not available in a non-Cre construct nor was the AAVrg-CAG-GFP virus available with a Cre insert. Addgene has now updated its inventory and the AAVrg pmSyn1-EBFP-Cre is no longer available and so we are unable to perform surgeries in WT animals at this time as suggested by the reviewers."

My response: Given that GFP expression has been demonstrated to cause toxicity, the concern is that any differences observed may be due to differences in viral transgene expression. Do AAVrg-CAG-GFP injected animals have same behavior as uninjected animals? At minimum, a strong emphasis must be placed on this caveat of the study. This is the only major concern remaining.

*Reviewer #3:*

The authors have addressed all my previous points satisfactorily. I do not have any further issues. The paper represents an important addition and tool to the field.

---

## [Author Response]

Essential revisions:The essential revisions are targeted at providing additional detail that was thought to be critical to the validity of the tool. Other details requested in the reviews are important, but these are particularly critical as they support the rationale for publishing. There are three:1) Correcting the emphasis on CRISPR as raised by reviewer 1.2) Immunostaining patterns comparing effects in males and females as raised by reviewer 2.3) Sequencing of PCR amplicons for the entire integrated transgene as raised by reviewer 1.

Reviewer #1:

The authors have developed a conditional knockout rat based on the Cre-loxP system to remove exon 3 of the Nr3c1 gene encoding the glucocorticoid receptor. They employ CRISPR/Cas9 in the development of a conditional knockout rat for the Nr3c1 gene. There is an overemphasis on the use of CRISPR/Cas9 starting with the title which is misleading in its current form. The conditional deletion of GR in rat brain is conditional on Cre not Cas9. Additionally, there is no evidence that CRISPR was required to make the animal. They obtained 1 animal out of 60 with the transgene. This could occur by injecting the targeting plasmid (as was done prior to the now common use of CRISPR for increased targeting). It is likely CRISPR aided in the process but there is no evidence for this (e.g. a plasmid alone was not injected and for good reason). The authors also do no appear to have done any PCR/Sequence/T7 assay/TIDE analysis on any of the 59 negative animals for evidence of CRISPR-induced mutation (which may have provided some possibly interesting GR mutants but this was not the goal). CRISPR should be deemphasized and focus on the fact there is a new Cre-dependent KO for GR and some possibly interesting biological observations using this new resource.

We appreciate the reviewer’s comments and have now changed the title to more accurately describe the content of the paper. We also wanted to clarify that in the first paragraph of the Results section, we described that we obtained 1 *Nr3c1* floxed animal out of 17 pups, as a result of microinjection of 60 fertilized eggs. This *Nr3c1* floxed animal was #60, which may have caused the viewer to think that 1 correctly targeted animal was obtained from 60. Therefore, we changed #60 to No. #60 in the text. In addition, the rest of 16 negative animals carried the alleles, including single loxP, large deletion between two sgRNA target sites, and indels at individual sgRNA target sites. As also mentioned by the reviewer, it was not our goal to study animals with a partially integrated transgene, indels at introns or knockout alleles (which is lethal) for the current study. Therefore, we do not show the genotyping data for the 16 negative animals in the manuscript.

The authors claim in the first paragraph of the Results section that offspring fecundity is evidence of no random integration. This is not the case. There may be random integration that doesn't affect breeding but disrupts other gene function. A copy number per genome determination is needed. A PCR assay internal to the transgene that uses one of the primers partially overlapping the loxP site to avoid amplification of endogenous locus and a PCR assay to a single copy gene in the genome (2 copies per genome) should be used for QPCR to determine transgene copy number per genome. This assay is becoming a standard practice in transgenic animal development. Also, if the entire plasmid was injected, PCR assays designed to detect other plasmid segments is recommended.

We fully agree with the reviewer’s comments and have presented the data showing the evidence of no random integration. We included two additional experiments in the revision. First, we performed a copy number analysis on three regions of the integrated area, including ‘5 to 5’ loxP, 3’ to 3’ loxP and between two loxP sites. We compared two WT and two homozygous *Nrc31* floxed rats and confirmed that there are 2 copies per genome for each of the three regions examined (Appendix 1—figure 2). Secondly, because a circular DNA plasmid was used for microinjection to reduce the random integration, we performed the PCR using the primers encompassing the either end of the plasmid backbone sequence. Both reactions did not detect any signals (Appendix 1—figure 2), consistent with the copy analysis indicating the absence of random integration. In addition, we would like to clarify that we described the fact “the offspring of rat #60 can be bred to homozygosity indicates the correct targeting of loxP sequences to the *Nr3c1* gene, instead of random integration”. We did not mention anywhere in the text that offspring fecundity is evidence of no random integration.

The authors do not appear to have sequenced PCR amplicons for the entire integrated transgene. This is important to show no mutations or aberrant repair occurred at the single nucleotide level. Years of work may be built on these animals that have a point mutation in exon 3 altering function of GR. Alignment of sequencing data to in silico predicted sequence should be provided as supplement.

We appreciate the reviewer’s suggestion. We have performed the long-range PCR using the primers encompassing the entire transgene region and sequenced from end to end. We confirmed that the DNA from two homozygous *Nr3c1* floxed animals showed no mutation in and near the coding sequence. However, we detected an A/T base change at chr18:31,744,353, compared to the rn6 reference genome, which is located in a non-conserved intronic region next to a SINE transposable element. Given that SD rats are an outbred strain and the location of the base change, we do not expect that this particular base difference will cause any significant effect. We have documented it in the revision (Results paragraph one).

The use pf AAVrg-CAG-GFP as control for AAVrg pmSyn1-EBFP-Cre is not clear. These are different promoters and transgenes (e.g. using a "dead" Cre in the same construct would be preferred). Consistent with other experiments, AAVrg pmSyn1-EBFP-Cre should be injected into WT animals as control.

The authors acknowledge that the use of these 2 viruses may not be entirely clear. The viruses used for these experiments were sourced from Addgene. At the time these experiments were performed, this combination of “control” and “Cre” viruses were the best available. The AAVrg pmSyn1-EBFP-Cre virus was not available in a non-Cre construct nor was the AAVrg-CAG-GFP virus available with a Cre insert. Addgene has now updated its inventory and the AAVrg pmSyn1-EBFP-Cre is no longer available and so we are unable to perform surgeries in WT animals at this time as suggested by the reviewers.

Overall, the finding of GR knockout in the prelimbic area affecting behavior is mostly confirmatory with published work while providing some interesting phenomenological observations for future studies. As such, this is a methods paper describing a transgenic rat with much potential and may be better in a specialized journal or methods section of neuroscience journal.

The conditional knockout of GR specifically in the CaMKIIα positive neurons in the prelimbic cortex, as opposed to the entire mPFC, is novel and has not been described elsewhere. The reviewer is correct in that, our lab and others have previously shown behavioral and hormonal consequences of mPFC GR knockdown; however, these studies did not include females, nor did they have a specific promotor driven knockdown (meaning all neurons were affected unlike our CAMKIIα selective neuronal population). Additionally, we and others have also employed Cre-Driver lines to knockout GR but these genetic models are confounded by the developmental and peripheral functions of GR. We believe the current dataset is a more targeted approach in terms of cellular specificity, brain region targeted and time constrained and adds to the growing body of literature. Further, to our knowledge, differences in diving behavior have not been published after GRKD manipulations.

Reviewer #2:

Scheimann et al. generated a conditional GR knockout in rats using CRISPR/Cas9 technology. They provide evidence of targeted GR deletion in specific cell types, brain regions, and circuits via viral vector-mediated delivery of Cre. They also report that deletion of GR in output neurons of the prelimbic cortex resulted in sex-specific deficits in extinction of conditioned fear in females and a shift to active coping in the forced swim test in males. The manuscript is well-written and development of a conditional GR knockout model in rat is important and will be useful tool for probing GR function in the brain. However, there are several major limitations to the data that should be addressed to strengthen the conclusions of the manuscript.

The authors appreciate this positive feedback and have addressed the specific concerns below.

1) Given the sex-specific effects on behavior, it is critical to show and evaluate the immunostaining patterns from both male and female rats. Are sex-specific differences observed in the distribution, efficiency, and/or extent of GR knockout in male and female brains?

These additional experiments have now been performed and added to the paper (Results subsection “Behavioral Consequences of Targeted GR Deletion; Implications for Fear and Coping Behaviors”; Figure 3; Materials and methods paragraph three of subsection “Immunohistochemistry”). We found significant knockdown in both females and males and no overt sex differences between groups.

2) Does the loss of GR lead to compensatory increases in the expression of the related receptor MR? This is an important consideration given that MR binds corticosterone, is co-expressed with GR in many brain regions, and is known to play a role in stress and emotional behavior.

We did not assess MR expression in the current study and although MR and GR both bind corticosterone we are particularly interested in the role of GR as it plays a stronger role in the “stress corticosterone state” and subsequent behavioral consequences such as fear and coping strategies. Although MR and GR are somewhat similar we do not believe that MR acts in a direct compensatory capacity in response to GR knockout. This is beyond the scope of the current work but may be addressed in future studies.

3) Several issues should be addressed concerning Figure 2:a) Why is the immunostaining pattern for GR in panel C different than in panel A? For the GR knockout cells in panel C, it appears that GR immunostaining is lost in the nucleus but not the cytoplasm. Why is this the case?

The immunostaining patterns appear different as 2 different GR antibodies were used. Panels A,B, E, F were all performed using a Santa Cruz (sc 1004) GR antibody prior to it being discontinued. Panels C and D were performed using an Invitrogen (PA1-511A) GR antibody. Both antibodies were rabbit polyclonal however they are not identical and thus have different immunostaining patterns.

b) The authors use AAVrg-CAG-GFP injection of SD:nr3c1fl/fl rats as a control for the AAVrg pmSyn1-EBFP-Cre injection (panels E and F). Another important control is injection of AAVrg pmSyn1-EBFP-Cre into wild-type rats. Was this control performed?

Addressed previously in response to reviewer #1.

“The authors acknowledge that the use of these 2 viruses may not be entirely clear. The viruses used for these experiments were sourced from Addgene. At the time these experiments were performed, this combination of “control” and “Cre” viruses were the best available. The AAVrg pmSyn1-EBFP-Cre virus was not available in a non-Cre construct nor was the AAVrg-CAG-GFP virus available with a Cre insert. Addgene has now updated its inventory and the AAVrg pmSyn1-EBFP-Cre is no longer available and so we are unable to perform surgeries in WT animals at this time as suggested by the reviewers.”

c) The images should be enlarged. The small size of the images makes it very difficult to evaluate the staining patterns.

All figures have now been uploaded as a separate files for clearer visualization.

4) Does knockout of GR in the prelimbic cortex alter locomotor activity of the rats?

We did not perform a separate measure of locomotion, such as the open field, in the current study.

5) The sample size for the GR knockout rats is quite small (n = 5 for females and n = 6 for males) for behavioral studies. In addition, a Fisher's LSD test was employed for post-hoc analyses. Are similar significance results found when statistical tests are utilized that correct for multiple comparisons?

We have re-ran our statistical analyses using Bonferroni post hoc to correct for multiple comparisons and found that all post hoc differences remain significant at p =/<0.05.

Reviewer #3:

This is the first study that develops a rat model with the capacity to perform cell- and region-specific deletion of the glucocorticoid receptor (GR) in rats. In addition to the novel tool development, the authors provided evidence of the behavioral impact of specific deletion of GR in the rat prelimbic CamKII-(glutamatergic) containing neurons. The data indicates differential effects of the knockdown approach on fear conditioning and extinction and behavior in the forced swim test in males and females, with a minor effect on corticosterone recovery in females. The study seems to be well performed and it is clearly written. Rats as model organisms have proved superior to mice for the dissection of stress (neuro)physiology and behavior and the current resource is a major step forward for the interrogation of GR function in this animal species.

The authors appreciate this positive feedback.

[Editors' note: further revisions were requested prior to acceptance, as described below.]The manuscript has been improved but there are some remaining issues that need to be addressed before acceptance, as outlined below:Please remove the statement concerning reviewer 1 below, and make certain the second issue raised is clearly noted as a caveat of the study if more data cannot be provided to address this directly.

Reviewer #1:

Below in quotes are the authors' responses to previous concerns. A follow up response is given. All other concerns were adequately addressed.Authors: "…addition, we would like to clarify that we described the fact "the offspring of rat #60 can be bred to homozygosity indicates the correct targeting of loxP sequences to the Nr3c1 gene, instead of random integration". We did not mention anywhere in the text that offspring fecundity is evidence of no random integration."My response: Although it is true the authors don't use the same verbiage, the ability to produce homozygous offspring should not be used as an indicator of correct targeting. The integration could still have additional integrants which would likely would be outcrossed if on a different chromosome or at least a sufficient distance from the targeted allele being genotyped. However, the only way to verify is to assay for such events and the authors have now done a more rigorous analysis to characterize the transgene integration. The statement in quotes provides an unnecessary precedent and contributes to the propagation of inadequate characterization of novel transgenic animals. It is not needed especially with the additional data and should be revised as simple statement of fact without indication or just remove it.

We appreciate the reviewer’s comment and have now changed the text accordingly to remove any indication. “We have consistently bred the offspring of rat #60 to homozygosity (Figure 1G).”

Authors: "The authors acknowledge that the use of these 2 viruses may not be entirely clear. The viruses used for these experiments were sourced from Addgene. At the time these experiments were performed, this combination of "control" and "Cre" viruses were the best available. The AAVrg pmSyn1-EBFP-Cre virus was not available in a non-Cre construct nor was the AAVrg-CAG-GFP virus available with a Cre insert. Addgene has now updated its inventory and the AAVrg pmSyn1-EBFP-Cre is no longer available and so we are unable to perform surgeries in WT animals at this time as suggested by the reviewers."My response: Given that GFP expression has been demonstrated to cause toxicity, the concern is that any differences observed may be due to differences in viral transgene expression. Do AAVrg-CAG-GFP injected animals have same behavior as uninjected animals? At minimum, a strong emphasis must be placed on this caveat of the study. This is the only major concern remaining.

We appreciate the reviewer’s comment and have now clearly acknowledged this caveat in the text (Results subsection “Validation of Conditional Gene Deletion: Viral Vector Targeting”). “Although we made every effort to employ appropriate viral controls, we were limited by the availability of viral constructs, specifically AAV retrograde constructs, and this is a caveat of the current study. For our intersectional approach we used the same AAV serotype (AAVrg) for both knockdown and controls however, the promoter (pmSyn1 vs CAG) and fluorescent reporter (EBFP vs GFP) were different. These constructs were chosen as the best available at the time of running the experiments.”

These specific animals were used for immunohistochemical analysis only and were not used in any behavioral testing so we are unable to say at this time if the GFP-virus led to changes in behavior.